# PRIMA: PLANNER-REASONER INSIDE A MULTI-TASK REASONING AGENT

## ABSTRACT

We consider the problem of multi-task reasoning (MTR), where an agent can solve multiple tasks via (first-order) logic reasoning. This capability is essential for human-like intelligence due to its strong generalizability and simplicity for handling multiple tasks. However, a major challenge in developing effective MTR is the intrinsic conflict between reasoning capability and efficiency. An MTR-capable agent must master a large set of "skills" to tackle diverse tasks, but executing a particular task at the inference stage requires only a small subset of immediately relevant skills. How can we maintain broad reasoning capability but efficient specific-task performance? To address this problem, we propose a Planner-Reasoner framework capable of state-of-the-art MTR capability and high efficiency. The Reasoner models shareable (first-order) logic deduction rules, from which the Planner selects a subset to compose into efficient reasoning paths. The entire model is trained in an end-to-end manner using deep reinforcement learning, and experimental studies over a variety of domains validate its effectiveness. [1]

## 1 INTRODUCTION

Multi-task learning (MTL) (Zhang & Yang, 2021; Zhou et al., 2011) demonstrates superior sample complexity and generalizability compared with the conventional "one model per task" style to solve multiple tasks. Recent research has additionally leveraged the great success of deep learning (LeCun et al., 2015) to empower learning deep multi-task models (Zhang & Yang, 2021; Crawshaw, 2020). Deep MTL models either learn a common multi-task feature representation by sharing several bottom layers of deep neural networks (Zhang et al., 2014; Liu et al., 2015; Zhang et al., 2015a; Mrksic et al., 2015; Li et al., 2015), or learn task-invariant and task-specific neural modules (Shinohara, 2016; Liu et al., 2017) via generative adversarial networks (Goodfellow et al., 2014). Although MTL is successful in many applications, a major challenge is the often impractically large MTL models. Although still smaller than piling up all models across different tasks, existing MTL models are significantly larger than a single model for tackling a specific task. This results from the intrinsic conflict underlying all MTL algorithms: balancing *across-task generalization capability* to perform different tasks with *single-task efficiency* in executing a specific task. On one hand, good generalization ability requires an MTL agent to be equipped with a large set of skills that can be combined to solve many different tasks. On the other hand, solving one particular task does not require all these skills. Instead, the agent needs to compose only a (small) subset of these skills into an efficient solution for a specific task. This conflict often hobbles existing MTL approaches.

This paper focuses on multi-task reasoning (MTR), a subarea of MTL that uses logic reasoning to solve multiple tasks. MTR is ubiquitous in human reasoning, where humans construct different reasoning paths for multiple tasks from the *same* set of reasoning skills. Conventional deep learning, although capable of strong expressive power, falls short in reasoning capabilities (Bengio, 2019). Considerable research has been devoted to endowing deep learning with logic reasoning abilities, the results of which include Deep Neural Reasoning (Jaeger, 2016), Neural Logic Reasoning (Besold et al., 2017; Bader et al., 2004; Bader & Hitzler, 2005), Neural Logic Machines (Dong et al., 2019), and other approaches (Besold et al., 2017; Bader et al., 2004; Bader & Hitzler, 2005). However, these approaches consider only single-task reasoning rather than a multi-task setting, and applying

---

[1]The code will be released after acceptance.

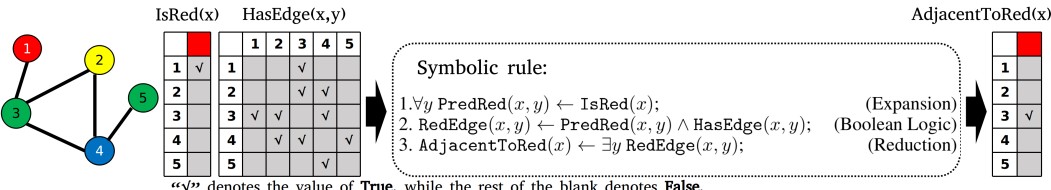

Figure 1: An example from the `AdjacentToRed` task and its formulation as a logical reasoning problem.

existing MTL approaches to learning these neural reasoning models leads to the same conflict between *across-task generalization capability* and *single-task efficiency*.

To strike a balance between reasoning capability and efficiency in MTR, we develop a ***Planner-Reasoner architecture Inside a Multi-task reasoning Agent*** (PRIMA) (Section 2), wherein the reasoner defines a set of neural logic operators for modeling reusable reasoning meta-rules ("skills") across tasks (Section 2.2). When defining the logic operators, we focus on first-order logic because of its simplicity and wide applicability to many reasoning problems, such as automated theorem proving (Fitting, 2012; Gallier, 2015) and knowledge-based systems (Van Harmelen et al., 2008). A separate planner module activates only a small subset of the meta-rules necessary for a given task and composes them into a deduction process (Section 2.3). Thus, our planner-reasoner architecture features the dual capabilities of *composing* and *pruning* a logic deduction process, achieving a graceful capability-efficiency trade-off in MTR (Section 2.4). The model architecture is trained in an end-to-end manner using deep reinforcement learning (Section 3), and experimental results on several benchmarks demonstrate that this framework leads to a more principled predicate space search and reduces reasoning complexity (Section 4). We discuss related works in Section 5, and conclude our paper in Section 6.

## 2 PLANNER-REASONER FOR MULTI-TASK REASONING

This section proposes the Planner-Reasoner framework for MTR. To that end, we first formally state the logic reasoning problem in Section 2.1, and then in Sections 2.2 and 2.3, we describe the ***Planner-Reasoner Inside a Single Reasoning Agent*** (PRISA) framework. This is a planner-reasoner architecture, which is a neural-logic architecture for traversing the first-order predicate space. Typically, logical reasoning can be reduced to *learning to search for a reasoning path with logical operators and then derive a logical consequence from premises*. Therefore, a reasoning problem can be addressed in two steps: (i) constructing the elementary logical operators and (ii) selecting a reasoning path that chains these logical operators together. This key observation motivates the Reasoner (Section 2.2) and Planner (Section 2.3) modules in our framework. In Section 2.4, the PRISA framework is extended to the MTR setting, which results in the ***Planner-Reasoner Inside a Multi-task Reasoning Agent*** (PRIMA) framework.

### 2.1 PROBLEM FORMULATION

**Logic reasoning** We begin with a brief introduction of the logic reasoning problem. Specifically, we consider a special variant of the First-Order Logic (FOL) system, which only consists of *individual variables*, *constants* and up to $r$-ary *predicate variables*. That is, we do not consider *functions* that map individual variables/constants into *terms*. An $r$-ary predicate $p(x_1, \ldots, x_r)$ can be considered as a relation between $r$ constants, which takes the value of `True` or `False`. An atom $p(x_1, \cdots, x_r)$ is an $r$-ary predicate with its arguments $x_1, \cdots, x_r$ being either variables or constants. A *well-defined formula* in our FOL system is a logical expression that is composed from atoms, logical connectives (e.g., negation $\neg$, conjunction $\wedge$, disjunction $\vee$, implication $\leftarrow$), and possibly existential $\exists$ and universal $\forall$ quantifiers according to certain formation rules (see Andrews (2002) for the details). In particular, the quantifiers $\exists$ and $\forall$ are only allowed to be applied to individual variables in FOL. In Fig. 1, we give an example from the `AdjacentToRed` task (Graves et al., 2016) and show how it could be formulated as a logical reasoning problem. Specifically, we are given a random graph along with the *properties* (i.e., the color) of the nodes and the *relations* (i.e., connectivity) between nodes. In our context, each node $i$ in the graph is a *constant* and an *individual variable* $x$ takes

values in the set of constants $\{1, \ldots, 5\}$. The properties of nodes and the relations between nodes are modeled as the unary predicate $\mathtt{IsRed}(x)$ ($5 \times 1$ vector) and the binary predicate $\mathtt{HasEdge}(x, y)$ ($5 \times 5$ matrix), respectively. The objective of logical reasoning is to deduce the value of the unary predicate $\mathtt{AdjacentToRed}(x)$ (i.e., whether a node $x$ has a neighbor of red color) from the *base* predicates $\mathtt{IsRed}(x)$ and $\mathtt{HasEdge}(x, y)$ (see Fig. 1 for an example of the deduction process).

**Multi-Task Reasoning**   Next, we introduce the definition of MTR. With a slight abuse of notations, let $\{p(x_1, \ldots, x_r) : r \in [1, n]\}$ be the set of input predicates sampled from any of the $k$ different reasoning tasks, where $x_1, \ldots, x_r$ are the individual variables and $n$ is the maximum arity. A multi-task reasoning model takes $p(x_1, \ldots, x_r)$ as its input and seeks to predict the corresponding ground-truth output predicates $q(x_1, \ldots, x_r)$. The aim is to learn multiple reasoning tasks jointly in a single model so that the reasoning skills in a task can be leveraged by other tasks to improve the general performance of all tasks at hand.

## 2.2 REASONER: TRANSFORMING LOGIC RULES INTO NEURAL OPERATORS

The Reasoner module conducts logical deduction using a set of neural operators constructed from first-order logic rules (more specifically, a set of "learnable" Horn clauses). Its architecture is inspired by NLM (Dong et al., 2019) (details about the difference can be found in Appendix D). Three logic rules are considered as essential meta-rules: $\mathtt{BooleanLogic}$, $\mathtt{Expansion}$, and $\mathtt{Reduction}$.

$$\mathtt{BooleanLogic}: \quad \mathtt{expression}(x_1, x_2, \cdots, x_r) \to \hat{p}(x_1, x_2, \cdots, x_r),$$

where $\mathtt{expression}$ is composed of a combination of Boolean operations ($\mathtt{AND}$, $\mathtt{OR}$, and $\mathtt{NOT}$) and $\hat{p}$ is the *output predicate*. For a given $r$-ary predicate and a given permutation $\psi \in S_n$, we define $p_\psi(x_1, \cdots, x_r) = p(x_{\psi(1)}, \cdots, x_{\psi(r)})$ where $S_n$ is the set of all possible permutations as the arguments to an input predicate. The corresponding neural implementation of $\mathtt{BooleanLogic}$ is $\sigma\left(\mathrm{MLP}\left(p_\psi(x_1, \cdots, x_r)\right); \theta\right)$, where $\sigma$ is the sigmoid activation function, MLP refers to a multi-layer perceptron, a $\mathrm{Permute}(\cdot)$ neural operator transforms input predicates to $p_\psi(x_1, \cdots, x_r)$, and $\theta$ is the learnable parameter within the model. This is similar to the implicit Horn clause with the universal quantifier($\forall$), e.g., $p_1(x) \wedge p_2(x) \to \hat{p}(x)$ implicitly denoting $\forall x \, p_1(x) \wedge p_2(x) \to \hat{p}(x)$. The class of neural operators can be viewed as "learnable" Horn clauses.

$\mathtt{Expansion}$, and $\mathtt{Reduction}$ are two types of meta-rules for quantification that bridge predicates of different arities with logic quantifiers ($\forall$ and $\exists$). $\mathtt{Expansion}$ introduces a new and distinct variable $x_{r+1}$ for a set of $r$-ary predicates with the universal quantifier($\forall$). For this reason, $\mathtt{Expansion}$ creates a new predicate $q$ from $p$.

$$\mathtt{Expansion}: \quad p(x_1, x_2, \cdots, x_r) \to \forall x_{r+1}, q(x_1, x_2, \cdots, x_r, x_{r+1}),$$

where $x_{r+1} \notin \{x_i\}_{i=1}^r$. The corresponding neural implementation of $\mathtt{Expansion}$, denoted by $\mathrm{Expand}(\cdot)$, expands the $r$-ary predicates into the $(r + 1)$-ary predicates by repeating the $r$-ary predicates and stacking them in a new dimension. Conversely, $\mathtt{Reduction}$ removes the variable $x_{r+1}$ in a set of $(r + 1)$-ary predicates via the quantifiers of $\forall$ or $\exists$.

$$\mathtt{Reduction}: \quad \forall x_{r+1} \, p(x_1, x_2, \cdots, x_r, x_{r+1}) \to q(x_1, x_2, \cdots, x_r), \text{or}$$
$$\exists x_{r+1} \, p(x_1, x_2, \cdots, x_r, x_{r+1}) \to q(x_1, x_2, \cdots, x_r).$$

The corresponding neural implementation of $\mathtt{Reduction}$, denoted by $\mathrm{Reduce}(\cdot)$, reducing the $(r+1)$-ary predicates into the $r$-ary predicates by taking the minimum (resp. maximum) along the dimension of $x_{r+1}$ due to the universal quantifier $\forall$ (resp. existential quantifier $\exists$).

## 2.3 PLANNER: ACTIVATING AND FORWARD-CHAINING LEARNABLE HORN CLAUSES

The Planner is our key module to address the capability-efficiency tradeoff in the MTR problem; it is responsible for activating the neural operators in the Reasoner and chaining them into reasoning paths. Existing learning-to-reason approaches, which are often based on inductive logic programming (ILP) (Cropper et al., 2020; Cropper & Dumancic, 2020; Muggleton & De Raedt, 1994) and the correspondingly neural-ILP methods (Dong et al., 2019; Shi et al., 2020). Conventional ILP methods suffer from several drawbacks, such as heavy reliance on human-crafted templates and sensitivity to noise. On the other hand, neural-ILP methods (Dong et al., 2019; Shi et al., 2020), leveraging

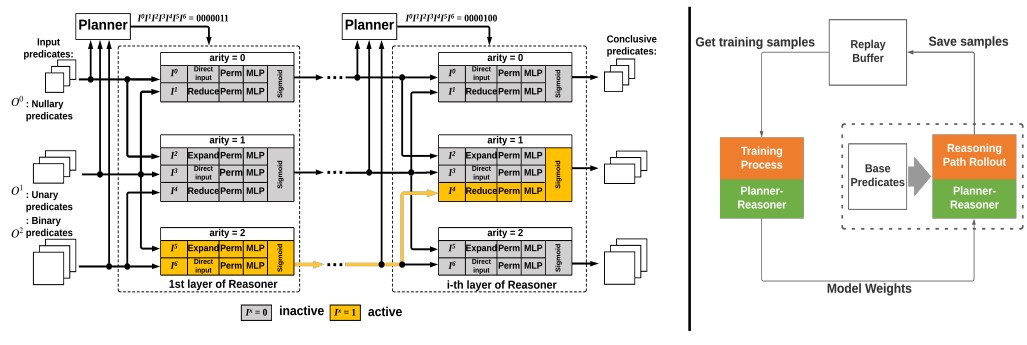

Figure 2: Left (A): The Planner-Reasoner Architecture with an example solution. Right (B): The overall (end-to-end) training process of the model by deep reinforcement learning. "Perm" denotes `Permute` operator.

the strength of deep learning and ILP, such as the Neural Logic Machine (NLM) (Dong et al., 2019), lack explicitness in the learning process of searching for reasoning paths. Let us take the learning process of NLM for example, which follows an intuitive two-step procedure. It first fully connects all the neural blocks and then searches all possible connections (corresponding to all possible predicate candidates) exhaustively to identify the desired reasoning path (corresponding to the desired predicate).

By using our proposed Planner module, we can strike a better capability-efficiency tradeoff. Rather than conducting an exhaustive search over all possible predicate candidates as in NLM, the Planner prunes all the unnecessary operators and identifies an essential reasoning path with low complexity for a given problem. Consider the following example. As shown in Fig. 1 and Fig. 2A (the highlighted orange-colored blocks), at each reasoning step, the Planner takes the input predicates and determines which neural operators should be activated. The decision is represented as a binary vector — $[I^0 \dots I^6]$ (termed *operator footprint*) — that corresponds to the neural operators of $\mathrm{Expand}$, $\mathrm{Reduce}$ and $\mathrm{DirectInput}$[2] at different arity groups. By chaining these sequential decisions, a sequence of operator footprints is formulated as a reasoning path, as the highlighted orange-colored path in Fig. 2A. Generally, the neural operators defined in the Reasoner can also be viewed as (learnable) Horn Clauses (Horn, 1951; Chandra & Harel, 1985), and the Planner *forward-chains* them into a reasoning path.

### 2.4 THE PRIMA FRAMEWORK FOR MULTI-TASK REASONING

By far, we have developed the PRISA architecture for the single-task reasoning setting. To extend PRISA to PRIMA for the multi-task reasoning setting, we add an extra nullary input predicate ($O^0$ in Fig. 2) to inform the planner of the current task. Based on the value of $O^0$, the planner will learn to activate the appropriate subset of neural operators, including a subset of shared operators and a few task-specific operators, for the current task. By doing so, PRIMA can learn to make the best use of reusable neural operators across different tasks. In Section 4, we will demonstrate such a capability of the planner on different tasks.

## 3 LEARNING-TO-REASON VIA REINFORCEMENT LEARNING

In our problem, the decision variables of the Planner are binary indicators of whether a neural operator module should be activated. Readers familiar with Markov decision processes (MDPs) (Puterman, 2014) might notice that the reasoning path of our model in Fig. 2A resembles a temporal rollout in MDP formulation (Sutton & Barto, 2018). Therefore, we frame learning-to-reason as a sequential decision-making problem and adopt off-the-shelf reinforcement learning (RL) algorithms.

---

[2]$\mathrm{DirectInput}$ is an identity mapping of the inputs where its following operators of $\mathrm{Permute}$, MLP and sigmoid nonlinearity can directly approximate the `BooleanLogic`.

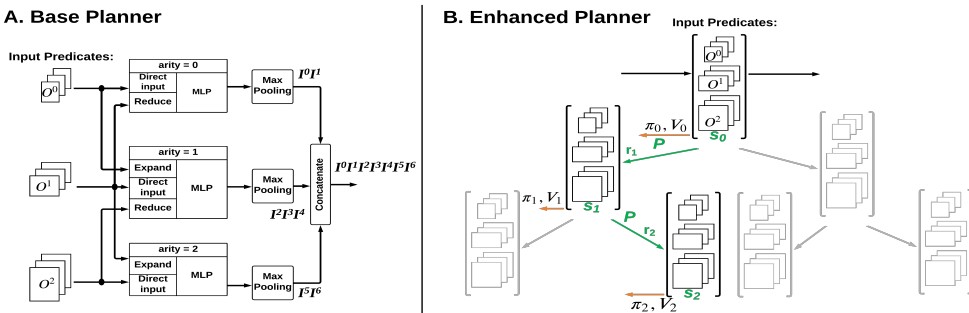

Figure 3: The architecture of the Base Planner and its enhanced version. Left: The Base Planner is based on a one-layer fully-activated Reasoner followed by max-pooling operations to predict an indicator of whether an op should be activated. Right: The Enhanced Planner uses MCTS to further boost the performance.

## 3.1 AN MDP FORMULATION OF THE LEARNING-TO-REASON PROBLEM

An MDP is defined as the tuple $(\mathcal{S}, \mathcal{A}, P_{ss'}^a, R, \gamma)$, where $\mathcal{S}$ and $\mathcal{A}$ are finite sets of states and actions, the transition kernel $P_{ss'}^a$ specifies the probability of transition from state $s \in \mathcal{S}$ to state $s' \in \mathcal{S}$ by taking action $a \in \mathcal{A}$, $R(s, a) : \mathcal{S} \times \mathcal{A} \to \mathbb{R}$ is the reward function, and $0 \le \gamma \le 1$ is a discount factor. A stationary policy $\pi : \mathcal{S} \times \mathcal{A} \to [0, 1]$ is a probabilistic mapping from states to actions. The primary objective of an RL algorithm is to identify a near-optimal policy $\pi^*$ that satisfies

$$\pi^* := \arg\max_\pi \left\{ J(\pi) := \mathbb{E}\left[ \sum_{t=0}^{T_{\max}-1} \gamma^t r(s_t, a_t \sim \pi) \right] \right\},$$

where $T_{\max}$ is a positive integer denoting the horizon length — that is, the maximum length of a rollout. The resemblance between Fig. 2 and an MDP formulation is relatively intuitive as summarized in Table 1. At the $t$-th time step, the state $s_t$ corresponds to the set of predicates, $s_t = [O_t^0, O_t^1, O_t^2]$, with the superscript denoting the corresponding arity. The action $a_t = [I_t^0, I_t^1, \ldots, I_t^{K-1}]$ (e.g., $a_0 = [000011]$) is a binary vector that indicates the activated neural operators (i.e., the operator footprint), where $K$ is the number of operators per layer. The reward $r_t$ is defined to be

$$r_t := \begin{cases} -\sum_{i=0}^{K-1} I_t^i, & \text{if } t < T_{\max} \\ \texttt{Accuracy}, & t = T_{\max} \end{cases}. \tag{1}$$

That is, the terminal reward is set to be the reasoning accuracy at the end of the reasoning path (see Appendix A.2 for its definition), and the intermediate reward at each step is chosen to be the negated number of activated operators (which penalizes the cost of performing the current reasoning step). The transition kernel $P_{ss'}^a$ corresponds to the function modeled by one Reasoner layer; each Reasoner layer will take the state (predicates) $s_{t-1} = [O_{t-1}^0, O_{t-1}^1, O_{t-1}^2,]$ and the action (operator footprint) $a_t = [I_t^0, I_t^1, \ldots, I_t^{K-1}]$ as its input and then generate the next state (predicates) $s_t = [O_t^0, O_t^1, O_t^2,]$. This also implies that the Reasoner layer defines a deterministic transition kernel, i.e., given $s_{t-1}$ and $a_t$ the next state $s_t$ is determined.

Table 1: The identification between the concepts of PRIMA/PRISA and that of RL at the $t$-th time step.

| RL | State $s_t$ | Action $a_t$ | Reward $r_t$ | Transition Kernel $P_{ss'}^a$ | Policy | Rollout |
|---|---|---|---|---|---|---|
| PRIMA/PRISA | Predicates of different arities: $[O_t^0, O_t^1, O_t^2]_t$ | Operator footprint: $[I_t^0 \ldots I_t^{K-1}]$ | Eq. (1) | One layer of **Reasoner** | **Planner** | Reasoning path |

## 3.2 POLICY NETWORK: MODELING OF PLANNER

The Planner module is embodied in the policy network. As shown in Fig. 3A, the base planner is a separate module that has the same architecture of 1-layer (fully-activated) Reasoner followed

by a max-pooling layer. This architecture enables the reduction of input predicates to the specific indicators by reflecting whether the operations at the corresponding position of one layer of Reasoner are active or inactive. Further, we can also leverage Monte-Carlo Tree Search (MCTS) (Browne et al., 2012; Munos, 2014) to boost the performance, which leads to an Enhanced Planner (Fig. 3B). An MCTS algorithm such as the Upper Confidence Bound for Trees (UCT) method (Kocsis et al., 2006), is a model-based RL algorithm that plans the best action at each time step (Browne et al., 2012) by constructing a search tree, with states as nodes and actions as edges. The Enhanced Planner uses MCTS to exploit partial knowledge of the problem structure (i.e., the deterministic transition kernel $P_{ss'}^a$ defined by the Reasoner layer) and construct a search tree to help identify the best actions (which ops to activate). Details of the MCTS algorithms used in the Enhanced Planner can be found in Appendix A.1.

### 3.3 OVERALL LEARNING FRAMEWORK

As illustrated in Fig. 3, we introduce concrete data-driven decision-making methods—that is, RL approaches (Sutton & Barto, 2018)—to address the learning-to-reason problem. To illustrate this, we apply the model-free RL method REINFORCE (Williams, 1992) and the model-based method MuZero (Schrittwieser et al., 2019). Compared with model-free reinforcement learning, model-based reinforcement learning (MBRL) more effectively handles large search-space problems, such as the game of Go (Silver et al., 2017b;b;a). MuZero (Schrittwieser et al., 2019), a recently proposed MBRL approach to integrating planning and learning, has achieved great success with a variety of complex tasks. Motivated by the success of MuZero, we propose an MBRL approach for neural-symbolic reasoning. The key insight behind adopting MuZero is that in real applications, we typically have partial structural knowledge of the transition kernel $P_{ss'}^a$ and reward function $r(s,a)$. As a result of the model-based module, testing complexity can be greatly reduced by adopting MCTS, which leads to a better set of predicates. Of course, MuZero is just one option in the model-based family of approaches. We leave it as future research to propose and compare other model-based alternatives.

The pipeline of the training process is illustrated in Fig 2.B (the right subfigure). After loading the model weights, the reasoning path rollouts are executed by the agent (or model instance), according to the current policy network. The performed reasoning path rollout is then stored in the replay buffer. The Planner-Reasoner is trained via rollouts sampled from the replay buffer.

## 4 EXPERIMENTAL RESULTS AND ANALYSIS

In this section, we evaluate the performance of different variants of PRISA (for single-task reasoning) and PRIMA (for multi-task reasoning) on eight tasks from the family tree and graph benchmarks (Graves et al., 2016), including `1-Outdegree`, `AdjacentToRed`, `HasFather`, `HasSister`, `4-Connectivity`, `IsGrandparent`, `IsUncle`, `IsMGUncle`. These tasks are widely used benchmark domains for inductive logic programming (Krötzsch, 2020; Calautti et al., 2015). Detailed descriptions about those tasks can be found in Appendix B.1. We evaluate their testing accuracy and reasoning cost (measured in FLOPs: the number of floating-point operations executed (Clark et al., 2020)) on these tasks and compare them to several baselines. Besides, the case study is conducted on the reasoning path, which indicates the operator sharing among different tasks. All the results demonstrate the graceful capability-efficiency tradeoff of PRIMA in multi-task reasoning.

### 4.1 EXPERIMENTAL SETUPS

Regarding the data generated in the single-task setting, we use the same methods as in NLM (Dong et al., 2019). A task will be randomly sampled in the multi-task setting according to a pre-defined probability distribution on different tasks. Compared to the generated data with that of the single task, it is augmented by using one-hot encoding for different tasks and wrapping it into nullary (background) predicates. Also, task-specific output heads are introduced to adapt to the multi-task setting. These adaptions apply to NLM-MTR, DLM-MTR, and PRIMA. The reasoning accuracy is used as the reward for both PRISA-MuZero and PRIMA. In the inference (or testing) stage, the Reasoner is combined with the learned Base Planner to perform the tasks for PRISA-MuZero and PRIMA, instead of an enhanced planner (MCTS), to reduce the extra computation. The problem

Table 2: **Testing Accuracy and PSS** of different variants of PRISA on different tasks. PRISA-MuZero achieves the best performance on single-task reasoning, which confirms the strength of the MCTS-based Enhanced Planner and the MuZero learning strategy. "*m*": the problem size. "PSS": Percentage of Successful Seeds.

|  |  |  | Family Tree | HasFather | HasSister | IsGrandparent | IsUncle | IsMGUncle | Graph | AdjacentToRed | 4-Connectivity | 1-OutDegree |
|---|---|---|---|---|---|---|---|---|---|---|---|---|
| testing acc | Single Task | PRISA-REINFORCE | m=20 | 62.6 | 50.7 | 96.5 | 97.3 | 99.8 | m=10 | 47.7 | 33.5 | 48.7 |
|  |  |  | m=100 | 87.8 | 69.8 | 2.3 | 97.7 | 98.4 | m=50 | 71.6 | 92.8 | 97.4 |
|  |  |  | PSS | 0 | 0 | 0 | 0 | 0 | PSS | 0 | 0 | 0 |
|  |  | PRISA-PPO | m=20 | 71.5 | 64.3 | 97.5 | 98.1 | 99.6 | m=10 | 62.3 | 57.8 | 61.6 |
|  |  |  | m=100 | 93.2 | 78.7 | 98.2 | 97.3 | 99.1 | m=50 | 85.5 | 95.2 | 96.3 |
|  |  |  | PSS | 0 | 0 | 0 | 0 | 0 | PSS | 0 | 0 | 0 |
|  |  | PRISA-MuZero | m=20 | **100** | **100** | **100** | **100** | **100** | m=10 | **100** | **100** | **100** |
|  |  |  | m=100 | **100** | **100** | **100** | **100** | **100** | m=50 | **100** | **100** | **100** |
|  |  |  | PSS | **100** | **100** | **100** | **100** | **100** | PSS | **90** | **100** | **100** |

size for training is always 10 for graph tasks and 20 for family tree tasks across all single-task and multi-task settings, regardless of the sizes of testing problems.

## 4.2 OVERALL PERFORMANCE

**Single-task reasoning capability**    First, we compare the performance of three different variants of PRISA (Section 3.3) for single-task reasoning, which learns their planners based on different reinforcement learning algorithms; PRISA-REINFORCE uses REINFORCE (Williams, 1992), PRISA-PPO uses PPO (Schulman et al., 2017), and PRISA-MuZero uses MuZero (Schrittwieser et al., 2019). We report the test accuracy and the Percentage of Successful Seeds (PSS) in Table 2 to measure the model's reasoning *capabilities*, where the PSS reaches 100% of success rates (Matthieu et al., 2021). We note that PRISA-MuZero has the same 100% accuracy as NLM (Dong et al., 2019), DLM (Matthieu et al., 2021), and $\partial$ILP (Evans & Grefenstette, 2018) across different tasks, and outperforms MemNN (Sukhbaatar et al., 2015) (shown in the single-task part in Table 3). But it also has a higher successful percentage (PSS) in comparison with other methods. The results show that PRISA-MuZero achieves the best performance on single-task reasoning, which confirms the strength of the MCTS-based Enhanced Planner (Section 3.3) and the MuZero learning strategy. Therefore, we will use the Enhanced Planner and MuZero in PRISA and PRIMA in the rest of our empirical studies.

**Multi-task reasoning capability**    Next, we evaluate the MTR capabilities of PRIMA. To the best of our knowledge, there is no existing approach that is designed specifically for MTR. Therefore, we adapt NLM and DLM into their multi-task versions, named NLM-MTR and DLM-MTR, respectively. NLM-MTR and DLM-MTR follow the same input and output modification as what we did to upgrade PRISA to PRIMA (Section 2.4). By this, we can examine the contribution of our proposed *Planner-Reasoner architecture* for MTR. As shown in Table 3, PRIMA (with MuZero as the Base Planner) performs well (perfectly) on different reasoning tasks. On the other hand, DLM-MTR experiences some performance degradation (on `AdjacentToRed`). This result confirms that our Planner-Reasoner architecture is more suitable for MTR. We conjecture that the benefit comes from using a Planner to explicitly select the necessary neural operators for each task, avoiding potential conflicts between different tasks during the learning process.

Experiments are also conducted to test the performance of PRIMA with different problem sizes. The problem size in training is 10 for all graph tasks and 20 for all family-tree tasks. In testing, we evaluate the methods on much larger problem sizes (50 for the graph tasks and 100 for the family tree tasks), which the methods have not seen before. Therefore, the Planner must dynamically activate a proper set of neural operators to construct a path to solve the new problem. As reported in Fig. 4 and Tables 2 and 3, PRIMA can achieve the best accuracy and lower flops when the problem sizes for training and testing are different.

**Reasoning efficiency**    To measure the reasoning efficiency of the proposed methods at the inference stage, PRIMA is compared with NLM, MemNN, and NLM-MTR in terms of FLOPs. As shown in Fig. 4, NLM and NLM-MTR demonstrate a similar performance and suffer the highest reasoning cost when it is tested with large problem sizes, such as 50 for graph tasks and 100 for family tree tasks. For MemNN, although the FLOPs of it seem low in most cases of testing, its testing accuracy is bad and cannot achieve accurate predictions (Table 3). In contrast, PRIMA can significantly reduce the reasoning complexity by intelligently selecting ops using a planner. Overall, PRIMA strikes a satisfactory capability-efficiency tradeoff in comparison with all available multi-tasking baselines.

Table 3: **Testing Accuracy and PSS** of PRIMA and other baselines on different reasoning tasks. The results of ∂ILP, NLM, and DLM are merged in one row due to space constraints and are presented in the same order if the results are different. Note that PRIMA's test efficiency is superior to NLM-MTR's as shown in Fig. 4. "*m*": the problem size. "PSS": Percentage of Successful Seeds. Numbers in red denote $< 100\%$.

| | | | Family Tree | HasFather | HasSister | IsGrandparent | IsUncle | IsMGUncle | Graph | AdjacentToRed | 4-Connectivity | 1-OutDegree |
|---|---|---|---|---|---|---|---|---|---|---|---|---|
| testing accuracy | Single Task | MemNN | m=20 | 99.9 | 86.3 | 96.5 | 96.3 | 99.7 | m=10 | 95.2 | 92.3 | 99.8 |
| | | | m=100 | 59.8 | 59.8 | 97.7 | 96 | 98.4 | m=50 | 93.1 | 81.3 | 78.6 |
| | | | PSS | 0 | 0 | 0 | 0 | 0 | PSS | 0 | 0 | 0 |
| | | ∂ILP/ NLM/ DLM | m=20 | 100 | 100 | 100 | 100 | 100 | m=10 | 100 | 100 | 100 |
| | | | m=100 | 100 | 100 | 100 | 100 | 100 | m=50 | 100 | 100 | 100 |
| | | | PSS | 100 | 100 | 100 | 100/ 90/ 100 | 100/ 20/ 70 | PSS | 100/ 90/ 90 | 100 | 100 |
| | Multi-Task | NLM-MTR | m=20 | 100 | 100 | 100 | 100 | 100 | m=10 | 100 | 100 | 100 |
| | | | m=100 | 100 | 100 | 100 | 100 | 100 | m=50 | 100 | 100 | 100 |
| | | | PSS | 100 | 100 | 100 | 100 | 90 | PSS | 90 | 100 | 100 |
| | | DLM-MTR | m=20 | 100 | 100 | 100 | 100 | 100 | m=10 | 96.7 | 100 | 100 |
| | | | m=100 | 100 | 100 | 100 | 100 | 100 | m=50 | 97.2 | 100 | 100 |
| | | | PSS | 100 | 100 | 100 | 100 | 100 | PSS | 0 | 100 | 100 |
| | | PRIMA | m=20 | 100 | 100 | 100 | 100 | 100 | m=10 | 100 | 100 | 100 |
| | | | m=100 | 100 | 100 | 100 | 100 | 100 | m=50 | 100 | 100 | 100 |
| | | | PSS | 100 | 100 | 100 | 100 | 90 | PSS | 90 | 100 | 100 |

**Operator/Path sharing in MTR** To take a closer look into how PRIMA achieves such a better capability-efficiency tradeoff (in Fig. 4), we examine the reasoning paths on three different graph tasks: `1-Outdegree`, `AdjacentToRed`, and `4-Connectivity`. Specifically, we sample instances from these three tasks and feed them into PRIMA separately to generate their corresponding reasoning paths. The results are plotted in Fig. 5A, where the gray paths denote the ones shared across tasks, and the colored ones are task-specific. It clearly shows that PRIMA learns a large set of neural operators sharable across tasks. Given each input instance from a particular task, PRIMA activates a set of shared paths along with a few task-specific paths to deduce the logical consequences.

**Generalizability of the Planner** To demonstrate the generalizability of the Planner module in PRIMA, we generate input instances of `AdjacentToRed` with topologies that have not been seen during training. In Fig. 5B, we show the reasoning paths activated by the Planner for these different topologies, which demonstrates that different input instances share a large portion of reasoning paths. This fact is not surprising as solving the same task of `AdjacentToRed` should rely on a common set of skills. More interestingly, we notice that even for solving this same task, the Planner will have to call upon a few *instance-dependent* sub-paths to handle the subtle inherent differences (e.g., the graph topology) that exist between different input instances. For this reason, PRIMA maintains an instance-dependent *dynamic* architecture, which is in sharp contrast to the Neural Architecture Search (NAS) approaches. Although NAS may also use RL algorithms to seek for a smaller architecture (Zoph & Le, 2017), it only searches for a *static* architecture that will be applied to all the input instances.

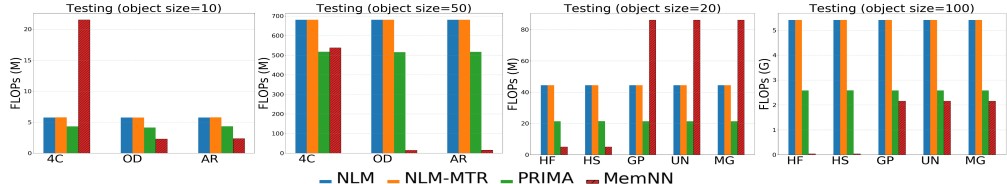

Figure 4: The reasoning costs (in FLOPs) of different models at the inference stage (1 M=$1\times10^6$, 1 G=$1\times10^9$). Compared to NLM and NLM-MTR, our PRIMA significantly reduces the reasoning complexity by intelligently selecting ops (short for neural operators) using a planner. Although the FLOPs of MemNN seem low in most cases of testing, its testing accuracy is bad and cannot achieve accurate predictions (see Table 3). "OD" denotes `1-Outdegree`, likewise, "AR":`AdjacentToRed`, "4C":`4-Connectivity`, "HF":`HasFather`, "HS":`HasSister`, "GP":`IsGrandparent`, "UN":`IsUncle`, "MG":`IsMGUncle`.

## 5 RELATED WORK

**Multi-task learning** Multi-task learning (Zhang & Yang, 2021; Zhou et al., 2011; Pan & Yang, 2009) has focused primarily on the supervised learning paradigm, which itself can be divided into several approaches. The first is feature-based MTL, also called multi-task feature learning, (Argyriou

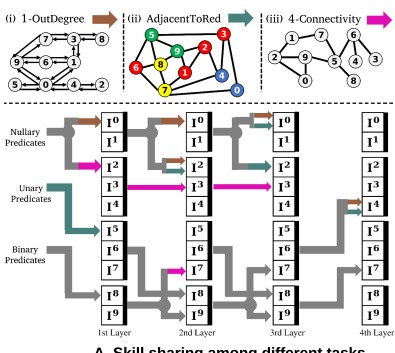 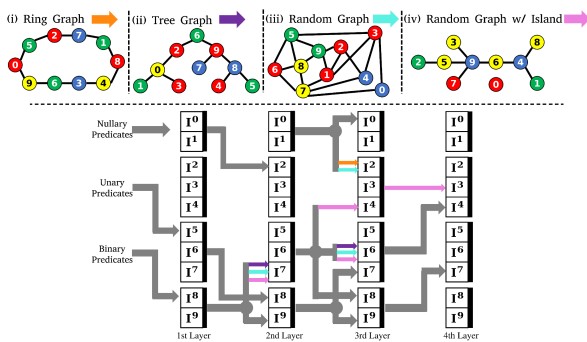

Figure 5: The reasoning paths of PRIMA. The gray arrows denote the "shared path". The colored arrows in sub-figures (A) and (B) denote task-specific paths and instance-specific paths, respectively. In (A), besides the AdjacentToRed task we have introduced earlier, we also consider the 1-Outdegree task, which reasons about whether the out-degree of a node is exactly equal to 1, and the 4-Connectivity task, which is to decide whether there are two nodes connected within 4 hops.

et al., 2008), which explores the sharing of features across different tasks using regularization techniques (Shinohara, 2016; Liu et al., 2017). The second approach assumes that tasks are intrinsically related, such as low-rank learning (Ando & Zhang, 2005; Zhang et al., 2005), learning with task clustering (Gu et al., 2011; Zhang et al., 2016), and task-relation learning (such as via task similarity, task correlation, or task covariance) (Goncalves et al., 2016; Zhang, 2013; Ciliberto et al., 2015). MTL has also been explored under other paradigms such as unsupervised learning, for example, via multi-task clustering in (Zhang & Zhang, 2013; Zhang et al., 2015b; Gu et al., 2011; Zhang, 2015).

**Neural-symbolic reasoning** Neural-symbolic AI for reasoning and inference has a long history (Besold et al., 2017; Bader et al., 2004; Garcez et al., 2008), and neural-ILP has developed primarily in two major directions (Cropper et al., 2020). The first direction applies neural operators such as tensor calculus to simulate logic reasoning (Yang et al., 2017; Dong et al., 2019; Shi et al., 2020) and (Manhaeve et al., 2018). This approach uses binary tensors over constant domains to represent the predicates and tensor chain products to simulate logic clauses. The second direction involves relaxed subset selection (Evans & Grefenstette, 2018; Si et al., 2019) with a predefined set of task-specific logic clauses. This approach reduces the task to a subset-selection problem by selecting a subset of clauses from the predefined set and using neural networks to search for a relaxed solution.

Our work is different from the most recent work, such as (Dong et al., 2019; Shi et al., 2020), in several ways. Compared with (Dong et al., 2019), the most notable difference is the improvement in efficiency by introducing learning-to-reason via reinforcement learning. A second major difference is that PRIMA offers more generalizability than NLM by decomposing the logic operators into more finely-grained units. We refer readers for more detailed related work to Appendix D.

## 6  CONCLUSION

A long-standing challenge in multi-task learning is the intrinsic conflict between capability and efficiency. In this paper, we propose a Planner-Reasoner framework for multi-task reasoning. To maintain broad capability but efficient specific performance, the Reasoner extracts shareable meta-rules via first-order logic, from which the Planner efficiently selects a subset of meta-rules to formulate the reasoning path. The model's training follows a complete data-driven end-to-end approach via deep reinforcement learning, and the performance is validated across a variety of benchmark tasks. Future work could include extending the framework to high-order logic and investigating scenarios when meta-rules have a hierarchical structure.

## REPRODUCIBILITY STATEMENT

We commit to ensuring that other researchers with reasonable background knowledge in our area can reproduce our theoretical and empirical results. The algorithm details can be seen in Appendix A. Specifically, benchmark details are provided in Appendix B.1, hyper-parameter settings are provided in Appendix B.2, and computing infrastructure are detailed in Appendix B.3. The experiment details can be seen in Appendix B.

## ETHICS STATEMENT

This work is about the methodology of achieving a capability-efficiency trade-off in multi-task first-order logic reasoning. The potential impact of this work is likely to further extend the framework to high-order logic and investigate scenarios when meta-rules have a hierarchical structure, which should be generally beneficial to the multi-task learning research community. We have not considered specific applications or practical scenarios as the goal of this work. Hence, it does not have any direct ethical consequences.

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

# SUPPLEMENTARY MATERIAL

## A ALGORITHM DETAILS

### A.1 DETAILS OF MCTS: FOUR KEY STEPS

During each epoch, MCTS repeatedly performs four sequential steps: selection, expansion, simulation, and backpropagation. The selection step traverses the existing search tree until the leaf node (or termination condition) by choosing actions (edges) $a_s$ at each node $s$ according to a tree policy. One widely used tree policy is the UCT (Kocsis et al., 2006) policy, which conducts the node-selection via

$$a_s = \arg\max_{s' \in \mathcal{C}(s)} \left\{ V_{s'} + \beta \sqrt{\frac{2 \log N_s}{N_{s'}}} \right\}, \tag{2}$$

where $\mathcal{C}(s)$ denotes the set of all child nodes for $s$, the first term $V_{s'}$ is an estimate of the long-term cumulative reward that can be received when starting from the state represented by node $s'$, and the second term represents the uncertainty (confidence-interval size) of that estimate. The confidence interval is calculated based on the upper confidence bound (UCB) (Auer et al., 2002; Auer, 2002) using $N_s$ and $N_{s'}$, which denote the number of times that nodes $s$ and $s'$ have been visited (respectively). The key idea of UCT policy (2) is to select the best action according to an optimistic estimation (the upper confidence bound) of the expected return, which balances the exploitation (first term) and exploration (second term) with $\beta$ controlling the trade-off. The second step (node expansion) is conducted according to a prior policy by adding a new child node if the selection process reaches a leaf node of the search tree. Next, the simulation step estimates the value function (cumulative reward) $\hat{V}_s$ by running the environment simulator with a default (simulation) policy. Finally, the *backpropagation* step updates the statistics $V_s$ and $N_s$ from the leaf node $s_T$ to the root node $s_0$ of the selected path by recursively performing the following update (i.e., from $t = T - 1$ to $t = 0$):

$$N_{s_t} \leftarrow N_{s_t} + 1, \quad \hat{V}_{s_t} \leftarrow r(s_t, a_t) + \gamma \hat{V}_{s_{t+1}}, \quad V_{s_t} \leftarrow \left( (N_{s_t} - 1) V_{s_t} + \hat{V}_{s_t} \right) / N_{s_t},$$

where $\hat{V}_{s_T}$ is the simulation return of $s_T$, and $a_t$ denotes the action selected following (2) at state $s_t$. Generally, the expansion step and the simulation step are more time-consuming than the other two steps, as they involve a large number of interactions with the environment.

In our Enhanced Planner, we adopt a recently proposed variant of MCTS that is based on probabilistic upper confidence tree (PUCT) (Rosin, 2011). It conducts node-selection according to a so-called PUCT score with two components $\text{PUCT}(s, a) = Q(s, a) + U(s, a)$. $Q(s, a)$ is the mean state-action value calculated from the averaged game result that takes action $a$ during current simulations, and $U(s, a)$ is the exploration bonus calculated as

$$U(s, a) = \pi(a|s) \frac{\sqrt{\sum_b N(s, b)}}{1 + N(s, a)} \left( c_1 + \log \left( \frac{\sum_b N(s, b) + c_2 + 1}{c_2} \right) \right),$$

where $a, b$ are possible actions, $\pi(a|s)$ is the output of the Base Planner, $N(s, a)$ is the visit count of the $(s, a)$ pair during current simulations, and constants $c_1, c_2$ are the exploration-controlling hyper-parameters.

An MCTS algorithm is a model-based RL algorithm that plans the best action at each time step (Browne et al., 2012) by constructing a search tree, with states as nodes and actions as edges. It uses the MDP model to identify the best action at each time step until the leaf node (or until other termination conditions are satisfied) by choosing actions (edges) $a_s$ at each node $s$ according to a tree policy. For implementations of Monte-Carlo Tree Search, we refer readers to `https://github.com/werner-duvaud/muzero-general` for details.

### A.2 DETAILS OF CALCULATING REASONING ACCURACY

Since the "target" predicates are available for all objects or pairs of objects, the reasoning accuracy refers to the accuracy evaluated on all objects (for properties such as `AdjacentToRed(x)` or `1-OutDegree(x)`) or pairs of objects (for relations such as `4-Connectivity(x, y)`).

# B    EXPERIMENTAL DETAILS

## B.1    REASONING TASKS INTRODUCTION

**Graph**    For graph tasks, they have the same background knowledge (or background predicate): `HasEdge`$(x, y)$, i.e., `HasEdge`$(x, y)$ is *True* if there is an undirected edge between node $x$ and node $y$. However, `AdjacentToRed` has an extra background predicate, `Red`$(x)$. `Red`$(x)$ is *True* if the color of node $x$ is red. Specifically, these graph tasks seeks to predict the target concepts (or target predicates) shown as below.

- `1-OutDegree`: `1-OutDegree`$(x)$ is *True* if the out-degree of node $x$ is exactly 1.

- `AdjacentToRed`: `AdjacentToRed`$(x)$ is *True* if the node $x$ has an edge with a red node.

- `4-Connectivity`: `4-Connectivity`$(x, y)$ is *True* if there are within 4 hops between node $x$ and node $y$.

**Family Tree**    For family tree tasks, they have the same background knowledge (or background predicates): `IsFather`$(x, y)$, `IsMother`$(x, y)$, `IsSon`$(x, y)$ and `IsDaughter`$(x, y)$. For instance, `IsFather`$(x, y)$ is *True* when $y$ is $x$'s father. Specifically, these family tree tasks seek to predict the target concepts (or target predicates) shown below.

- `HasFather`: `HasFather`$(x)$ is *True* if $x$ has father.

- `HasSister`: `HasSister`$(x)$ is *True* if $x$ has at least one sister.

- `IsGrandparent`: `IsGrandparent`$(x, y)$ is *True* if $y$ is $x$'s grandparent.

- `IsUncle`: `IsUncle`$(x, y)$ is *True* if $y$ is $x$'s uncle.

- `IsMGUncle`: `IsMGUncle`$(x, y)$ is *True* if $y$ is $x$'s maternal great uncle.

## B.2    HYPER-PARAMETER SETTINGS

For simplicity, we set discount factor $\gamma = 1$ in all experiments. The $T_{\max}$ depends on the depth of the Reasoner. In single-task setting, we use the same hyper-parameter settings as in the original papers for MemNN (Sukhbaatar et al., 2015), $\partial$ILP (Evans & Grefenstette, 2018), NLM (Dong et al., 2019), and DLM (Matthieu et al., 2021). In multi-task setting, the probability distribution of sampling a task for NLM-MTR is $[0.09, 0.11, 0.11, 0.09, 0.09, 0.135, 0.135, 0.24]$ that is corresponding to the task list of [`1-Outdegree`, `AdjacentToRed`, `4-Connectivity`, `HasFather`, `HasSister`, `IsGrandparent`, `IsUncle`, `IsMGUncle`]. That distribution for DLM-MTR is $[0.1, 0.12, 0.12, 0.1, 0.1, 0.13, 0.13, 0.2]$. The details of hyper-parameters in NLM-MTR and DML-MTR can be found in Table 4. For all MLPs inside NLM-MTR, we keep the same settings as NLM. Similarly, the "Internal Logic", initial temperature and scale of the Gumbel distribution, and the dropout probability in DLM-MTR are also kept the same settings as those in DLM. Besides, the probability distribution of sampling a task for PRIMA is $[0.1, 0.12, 0.12, 0.1, 0.1, 0.13, 0.13, 0.2]$. Regarding the hyper-parameters of PRIMA, details can be found in Table 5. We also show the hyper-parameters of PRISA-REINFORCE, PRISA-PPO, PRISA-MuZero in Table 5.

We note that the problem size for training is always 10 for graph tasks and 20 for family tree tasks in single-task and multi-task settings.

## B.3    COMPUTING INFRASTRUCTURE

We conducted our experiments on a CPU server where the CPU is "Intel(R) Xeon(R) Silver 4114 CPU" with 40 cores and 64 GB memory in total.

Table 4: Hyper-parameter settings for NLM-MTR and DML-MTR.

|  | NLM-MTR | DML-MTR |
|---|---|---|
| learning rate | 0.005 | 0.005 |
| epochs | 50 | 200 |
| epoch size | 8000 | 2000 |
| batch size (train) | 4 | 4 |
| breadth | 3 | 3 |
| depth | 4 | 9 |

Table 5: Hyper-parameter settings for PRISA and PRIMA. "lr-reasoner" refers to the learning rate for the reasoner, "lr-policy" denotes the learning rate for the policy network, "lr-value" denotes the learning rate for the value network, "residual" refers to the residual connection in the reasoner, "NumWarmups" refers to the number of warm-ups before starting the training, "NumRollouts" refers to the number of roll-outs in MCTS, "RwdDecay" refers to the constant exponential decay applied on the reward, "$c_1$ and $c_2$" refers to the same constants in PUCT formula in Section A.1, "RBsize" refers to the replay buffer size that is used to count the number of stored trajectories, "BatchSize" refers to the batch size for training, "TrainingSteps" refers to the number of training steps. For PRISA-REINFORCE, PRISA-PPO, PRISA-MuZero, the "TrainingSteps" denotes the training steps for each task, while that of PRIMA represents the training steps for all 8 tasks.

|  | PRISA-REINFORCE | PRISA-PPO | PRISA-MuZero | PRIMA |
|---|---|---|---|---|
| lr-reasoner | 0.005 | 0.005 | 0.005 | 0.004 |
| lr-policy | 0.085 | 0.085 | 0.075 | 0.075 |
| lr-value | - | 0.15 | 0.075 | 0.075 |
| breadth | 3 | 3 | 3 | 3 |
| depth | 4 | 4 | 4 | 4 |
| residual | *False* | *False* | *False* | *False* |
| NumWarmups | - | - | 200 | 200 |
| NumRollouts | - | - | 1200 | 1200 |
| RwdDecay | 5 | 5 | 5 | 5 |
| $c_1$ | - | - | 30 | 30 |
| $c_2$ | - | - | 19652 | 19652 |
| RBsize | 16 | 32 | 400 | 400 |
| BatchSize | 16 | 16 | 16 | 16 |
| TrainingSteps | $7 \times 10^4$ | $7 \times 10^4$ | $7 \times 10^4$ | $39 \times 10^4$ |

Table 6: Additional results: The performance of PRIMA w.r.t different numbers of intermediate predicates. "# pred": the number of intermediate predicates.

| | | Family Tree | HasFather | HasSister | IsGrandparent | IsUncle | IsMGUncle | Graph | AdjacentToRed | 4-Connectivity | 1-OutDegree |
|---|---|---|---|---|---|---|---|---|---|---|---|
| testing accuracy | # pred = 2 | m=20 | 61.78 | 54.97 | 96.46 | 97.19 | 99.67 | m=10 | 42.62 | 89.28 | 81.88 |
| | | m=100 | 87.5 | 31.2 | 97.7 | 96.5 | 98.4 | m=50 | 11.1 | 96.6 | 96.5 |
| | | PSS | 0 | 0 | 0 | 0 | 0 | PSS | 0 | 0 | 0 |
| | # pred = 4 | m=20 | 100 | 100 | 100 | 97.2 | 99.7 | m=10 | 67.3 | 93.4 | 100 |
| | | m=100 | 100 | 100 | 100 | 96.6 | 98.4 | m=50 | 91.7 | 97.8 | 100 |
| | | PSS | 100 | 100 | 100 | 0 | 0 | PSS | 0 | 0 | 100 |
| | # pred = 6 | m=20 | 100 | 100 | 100 | 100 | 100 | m=10 | 99.7 | 100 | 100 |
| | | m=100 | 100 | 100 | 100 | 100 | 100 | m=50 | 100 | 100 | 100 |
| | | PSS | 100 | 100 | 100 | 100 | 90 | PSS | 0 | 100 | 100 |
| | # pred = 8 | m=20 | 100 | 100 | 100 | 100 | 100 | m=10 | 100 | 100 | 100 |
| | | m=100 | 100 | 100 | 100 | 100 | 100 | m=50 | 100 | 100 | 100 |
| | | PSS | 100 | 100 | 100 | 100 | 90 | PSS | 90 | 100 | 100 |

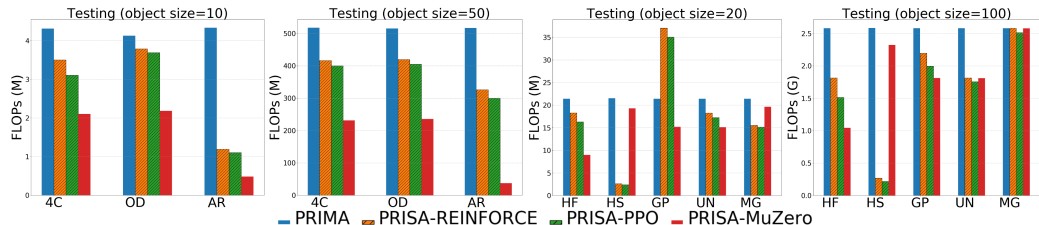

Figure 6: The reasoning costs (in FLOPs) of PRIMA and three variants of PRISA at the inference stage (1 M=$1\times10^6$, 1 G=$1\times10^9$).

## C ADDITIONAL EXPERIMENTS

### C.1 REASONING COSTS OF PRISA

To demonstrate the improved reasoning efficiency in PRISA at the inference stage, three variants of PRISA (e.g., PRISA-REINFORCE, PRISA-PPO, and PRISA-MuZero) are compared with PRIMA in terms of FLOPs. As shown in Fig. 6, three variants of PRISA generally have lower reasoning complexity than PRIMA. The difference in FLOPs between PRIMA and PRISA-MuZero is primarily from the difference between multi-tasking setting and single-task setting.

### C.2 PREDICATE DIMENSION ANALYSIS

Since the Reasoner only realizes a partial set of Horn clauses of FOL, the number of intermediate predicates will greatly determine the expressive power of the model. The performance of PRIMA degrades gracefully when the number of intermediate predicates decreases. This is shown in the additional experiments of examining the limiting performance of our PRIMA, where we examine the limiting performance of our PRIMA when the number of intermediate predicates at each layer decreases. The results in Table 6 show that the performance of PRIMA degrades gracefully when the number of intermediate predicates decreases; for example, it still performs reasonably well on most tasks even when the number of predicates becomes 4.

### C.3 ANALYSIS OF MULTI-TASKING CAPABILITIES

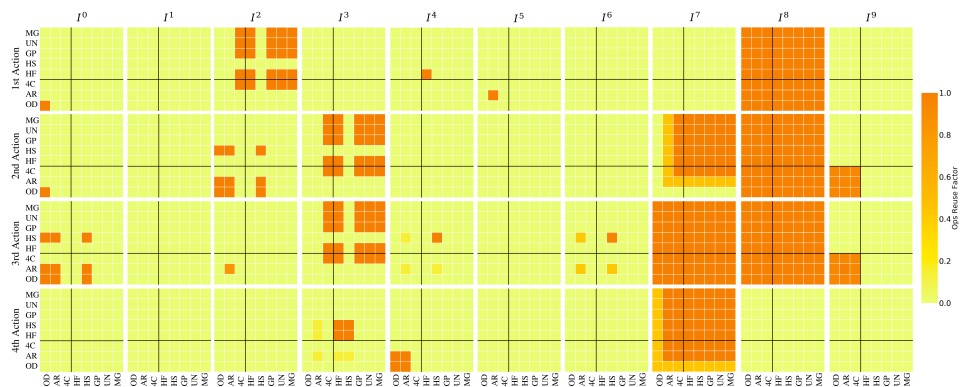

Figure 7: The quantitative evaluation of ops reuse between different Tasks in PRIMA.

**Why/how do different tasks share blocks (of ops) as in Fig. 7?** To quantitatively measure how different tasks reuses the "skills", we show in Fig. 7, which is a heat-map showing the probability of overlapping about Ops reuse across different tasks, where red color denotes a high-level of Ops overlapping. Ops reuse factor is defined as $\min(P_a(I^j), P_b(I^j))$ where $P_a(I^j)$ refers to the probability of ops $I^j (j \in \{0, \ldots, 9\})$ being active for task $a$ and $P_b(I^j)$ denotes the probability of $I^j$ being active for task $b$. From Fig. 7, it can be inferred that some tasks share skills more than others. For example, in most of the actions, $I^8$ is shared across different tasks.

# D  ADDITIONAL RELATED WORK

This section provides complementary contents to Sec. 5.

**Model-based symbolic planning**  Another large body of related work is symbolic planning (SP) (Van Harmelen et al., 2008; Hanheide et al., 2015; Chen et al., 2016; Khandelwal et al., 2017). In SP approaches, a planning agent carries prior symbolic knowledge of objects, properties, and how they are changed by executing actions in the dynamic system, represented in a formal, logic-based language such as PDDL (McDermott et al., 1998) or an action language (Gelfond & Lifschitz, 1998) that relates to logic programming under answer set semantics (answer set programming) (Lifschitz, 2008). The agent utilizes a symbolic planner, such as a PDDL planner FASTDOWNWARD (Helmert, 2006) or an answer set solver CLINGO (Gebser et al., 2012) to generate a sequence of actions based on its symbolic knowledge, executes the actions to achieve its goal. Compared with RL approaches, an SP agent does not require a large number of trial-and-error to behave reasonably well, yet requires predefined symbolic knowledge as the model prior. It should be noted that it remains questionable if PDDL-based methods can be applied directly to our problem, as there is no explicitly predefined prior knowledge under our problem setting, which is required by PDDL-based approaches. In contrast, we address the learning-to-reason problem via a complete data-driven deep reinforcement approach.

**Multi-task RL**  Multi-task learning can help boost the performance of reinforcement learning, leading to multi-task reinforcement learning (Vithayathil Varghese & Mahmoud, 2020; Zhu et al., 2020). Some research (Wilson et al., 2007; Li et al., 2009; Lazaric & Ghavamzadeh, 2010; Calandriello et al., 2014; Andreas et al., 2017; Deshmukh et al., 2017; Saxe et al., 2017; Bräm et al., 2019; Vuong et al., 2019; Igl et al., 2020) has adapted the ideas behind multi-task learning to RL. For example, in (Bräm et al., 2019), a multi-task deep RL model based on attention can group tasks into sub-networks with state-level granularity. The idea of compression and distillation has been incorporated into multi-task RL as in (Parisotto et al., 2016; Rusu et al., 2016; Omidshafiei et al., 2017; Teh et al., 2017). For example, in (Parisotto et al., 2016), the proposed actor-mimic method combines deep reinforcement learning with model-compression techniques to train a policy network that can learn to act for multiple tasks. Other research in multi-task RL focuses on online and distributed settings (Bsat et al., 2017; Sharma & Ravindran, 2017; Sharma et al., 2018; Espeholt et al., 2018; Tutunov et al., 2018; Lin et al., 2019).

**RL for logic reasoning**  Relational RL integrates RL with statistical relational learning and connects RL with classical AI for knowledge representation and reasoning. Most prominently, Džeroski et al. (2001) originally proposed relational RL, Tadepalli et al. (2004) surveyed relational RL, Guestrin et al. (2003) introduced relational MDPs, and Diuk et al. (2008) introduced objected-oriented MDPs (OO-MDPs). More recently, Battaglia et al. (2018) proposed to incorporate relational inductive bias, Zambaldi et al. (2018) proposed deep relational RL, Keramati et al. (2018) proposed strategic object-oriented RL, and some researchers have also adopted deep learning approaches for dealing with relations and/or reasoning, for example (Battaglia et al., 2016; Chen et al., 2018; Santoro et al., 2017; Santurkar et al., 2018; Palm et al., 2017; Xia et al., 2018; Yi et al., 2018).

**Statistical relational reasoning**  Statistical relational reasoning (SRL) (Koller et al., 2007) often provides a better understanding of domains and predictive accuracy, but with more complex learning and inference processes. Unlike ILP, SRL takes a statistical and probabilistic learning perspective and extends the probabilistic formalism with relational aspects. Examples of SRL include text classification (Ganiz et al., 2010), recommendation systems (Cao, 2016), and wireless networks (Yoshida et al., 2020). SRL can be divided into two research branches: probabilistic relational models (PRMs) (Koller, 1999) and probabilistic logic models (PLMs) (Chen et al., 2008). PRMs start from probabilistic graphical models and then extend to relational aspects, while PLMs start from ILP and extend to probabilistic semantics. A related research area is graph relational reasoning, such as using graph neural networks (Zhou et al., 2020) to conduct reasoning. A discussion of this approach is beyond the scope of this paper, but we refer interested readers to (Chakrabarti & Faloutsos, 2006; Cook & Holder, 2006; Zhang et al., 2019) for a comprehensive review.

**Comparisons with NLM in details**  The Reasoner shares the same functionality with NLM, in terms of approximating the meta-rules (`BooleanLogic`, `Expansion`, and `Reduction`). However,

our Reasoner has major differences from NLM. *First*, the targets are different. Our Reasoner targets the trade-off between capability and efficiency. Compared with NLM, our Reasoner can greatly improve the efficiency by using a Planner module. *Second*, our Reasoner has more explicitness in the reasoning path. Let $[O_{i-1}^{r-1}, O_{i-1}^{r}, O_{i-1}^{r+1}]$ denote the output predicates (in tensor representation) from the previous layer, and $\mathrm{Concate}(\cdot)$ denotes the concatenation operation. NLM performs the intra-group computation as

$$O_i^r = \sigma\left(\mathrm{MLP}\left(\mathrm{Permute}\left(Z_i^r\right); \theta_i^r\right)\right),$$

where $O_i^r$ is the output predicate, $Z_i^r = \mathrm{Concat}\left(\mathrm{Expand}\left(O_{i-1}^{r-1}\right), O_{i-1}^r, \mathrm{Reduce}\left(O_{i-1}^{r+1}\right)\right)$, $\sigma$ is the sigmoid nonlinearity and $\theta_i^r$ denotes learnable parameters. On the contrary, in the Reasoner module of our PRISA/PRIMA, the implementation is

$$O_i^r = \sigma\Bigg(\mathrm{MLP}\big(\mathrm{Permute}\big(\mathrm{Expand}(O_{i-1}^{r-1})\big); \theta_i^r\big) + \mathrm{MLP}\big(\mathrm{Permute}\big(O_{i-1}^r\big); \theta_i^r\big)$$
$$+ \mathrm{MLP}\big(\mathrm{Permute}\big(\mathrm{Reduce}(O_{i-1}^{r+1})\big); \theta_i^r\big)\Bigg),$$

which allows for the neural implementations of `BooleanLogic`, `Expansion`, and `Reduction` (at the same arity group) to be executed independently to some extent and reduce the unnecessary computations.

# PRIMA: PLANNER-REASONER INSIDE A MULTI-TASK REASONING AGENT

**Anonymous authors**

## ABSTRACT

We consider the problem of multi-task reasoning (MTR), where an agent can solve multiple tasks via (first-order) logic reasoning. This capability is essential for human-like intelligence due to its strong generalizability and simplicity for handling multiple tasks. However, a major challenge in developing effective MTR is the intrinsic conflict between reasoning capability and efficiency. An MTR-capable agent must master a large set of "skills" to tackle diverse tasks, but executing a particular task at the inference stage requires only a small subset of immediately relevant skills. How can we maintain broad reasoning capability but efficient specific-task performance? To address this problem, we propose a Planner-Reasoner framework capable of state-of-the-art MTR capability and high efficiency. The Reasoner models shareable (first-order) logic deduction rules, from which the Planner selects a subset to compose into efficient reasoning paths. The entire model is trained in an end-to-end manner using deep reinforcement learning, and experimental studies over a variety of domains validate its effectiveness.

## 1 INTRODUCTION

Multi-task learning (MTL) (Zhang & Yang, 2021; Zhou et al., 2011) demonstrates superior sample complexity and generalizability compared with the conventional "one model per task" style to solve multiple tasks. Recent research has additionally leveraged the great success of deep learning (LeCun et al., 2015) to empower learning deep multi-task models (Zhang & Yang, 2021; Crawshaw, 2020). Deep MTL models either learn a common multi-task feature representation by sharing several bottom layers of deep neural networks (Zhang et al., 2014; Liu et al., 2015; Zhang et al., 2015a; Mrksic et al., 2015; Li et al., 2015), or learn task-invariant and task-specific neural modules (Shinohara, 2016; Liu et al., 2017) via generative adversarial networks (Goodfellow et al., 2014). Although MTL is successful in many applications, a major challenge is the often impractically large MTL models. Although still smaller than piling up all models across different tasks, existing MTL models are significantly larger than a single model for tackling a specific task. This results from the intrinsic conflict underlying all MTL algorithms: balancing *across-task generalization capability* to perform different tasks with *single-task efficiency* in executing a specific task. On one hand, good generalization ability requires an MTL agent to be equipped with a large set of skills that can be combined to solve many different tasks. On the other hand, solving one particular task does not require all these skills. Instead, the agent needs to compose only a (small) subset of these skills into an efficient solution for a specific task. This conflict often hobbles existing MTL approaches.

This paper focuses on multi-task reasoning (MTR), a subarea of MTL that uses logic reasoning to solve multiple tasks. MTR is ubiquitous in human reasoning, where humans construct different reasoning paths for multiple tasks from the *same* set of reasoning skills. Conventional deep learning, although capable of strong expressive power, falls short in reasoning capabilities (Bengio, 2019). Considerable research has been devoted to endowing deep learning with logic reasoning abilities, the results of which include Deep Neural Reasoning (Jaeger, 2016), Neural Logic Reasoning (Besold et al., 2017; Bader et al., 2004; Bader & Hitzler, 2005), Neural Logic Machines (Dong et al., 2019), and other approaches (Besold et al., 2017; Bader et al., 2004; Bader & Hitzler, 2005). However, these approaches consider only single-task reasoning rather than a multi-task setting, and applying existing MTL approaches to learning these neural reasoning models leads to the same conflict between *across-task generalization capability* and *single-task efficiency*.

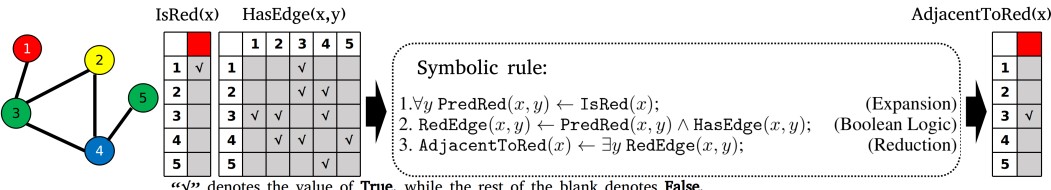

Figure 1: An example from the `AdjacentToRed` task and its formulation as a logical reasoning problem.

To strike a balance between reasoning capability and efficiency in MTR, we develop a ***Planner-Reasoner architecture Inside a Multi-task reasoning Agent*** (PRIMA) (Section 3), wherein the reasoner defines a set of neural logic operators for modeling reusable reasoning meta-rules ("skills") across tasks (Section 3.2). When defining the logic operators, we focus on first-order logic because of its simplicity and wide applicability to many reasoning problems, such as automated theorem proving (Fitting, 2012; Gallier, 2015) and knowledge-based systems (Van Harmelen et al., 2008). A separate planner module activates only a small subset of the meta-rules necessary for a given task and composes them into a deduction process (Section 3.3). Thus, our planner-reasoner architecture features the dual capabilities of *composing* and gpruning a logic deduction process, achieving a graceful capability-efficiency trade-off in MTR. The model architecture is trained in an end-to-end manner using deep reinforcement learning (Section 4), and experimental results on several benchmarks demonstrate that this framework leads to a state-of-the-art balance between capability and efficency (Section 5). We discuss related works in Section 6, and conclude our paper in Section 7.

## 2    PROBLEM FORMULATION

**Logic reasoning**    We begin with a brief introduction of the logic reasoning problem. Specifically, we consider a special variant of the First-Order Logic (FOL) system, which only consists of *individual variables*, *constants* and up to $r$-ary *predicate variables*. That is, we do not consider *functions* that map individual variables/constants into *terms*. An $r$-ary predicate $p(x_1, \ldots, x_r)$ can be considered as a relation between $r$ constants, which takes the value of `True` or `False`. An atom $p(x_1, \cdots, x_r)$ is an $r$-ary predicate with its arguments $x_1, \cdots, x_r$ being either variables or constants. A *well-defined formula* in our FOL system is a logical expression that is composed from atoms, logical connectives (e.g., negation $\neg$, conjunction $\wedge$, disjunction $\vee$, implication $\leftarrow$), and possibly existential $\exists$ and universal $\forall$ quantifiers according to certain formation rules (see Andrews (2002) for the details). In particular, the quantifiers $\exists$ and $\forall$ are only allowed to be applied to individual variables in FOL. In Fig. 1, we give an example from the `AdjacentToRed` task (Graves et al., 2016) and show how it could be formulated as a logical reasoning problem. Specifically, we are given a random graph along with the *properties* (i.e., the color) of the nodes and the *relations* (i.e., connectivity) between nodes. In our context, each node $i$ in the graph is a *constant* and an *individual variable* $x$ takes values in the set of constants $\{1, \ldots, 5\}$. The properties of nodes and the relations between nodes are modeled as the unary predicate `IsRed`$(x)$ ($5 \times 1$ vector) and the binary predicate `HasEdge`$(x, y)$ ($5 \times 5$ matrix), respectively. The objective of logical reasoning is to deduce the value of the unary predicate `AdjacentToRed`$(x)$ (i.e., whether a node $x$ has a neighbor of red color) from the *base* predicates `IsRed`$(x)$ and `HasEdge`$(x, y)$ (see Fig. 1 for an example of the deduction process).

**Multi-task reasoning**    Next, we introduce the definition of MTR. With a slight abuse of notations, let $\{p(x_1, \ldots, x_r) : r \in [1, n]\}$ be the set of input predicates sampled from any of the $k$ different reasoning tasks, where $x_1, \ldots, x_r$ are the individual variables and $n$ is the maximum arity. A multi-task reasoning model takes $p(x_1, \ldots, x_r)$ as its input and seeks to predict the corresponding ground-truth output predicates $q(x_1, \ldots, x_r)$. The aim is to learn multiple reasoning tasks jointly in a single model so that the reasoning skills in a task can be leveraged by other tasks to improve the general performance of all tasks at hand.

## 3    PRIMA: PLANNER-REASONER FOR MULTI-TASK REASONING

In this section, we develop our PRIMA framework, which is a novel neural-logic model architecture for multi-task reasoning. We begin with an introduction to the overall architecture (Section 3.1) along with its key design insights, and then proceed to discuss its two modules in Sections 3.2–3.3.

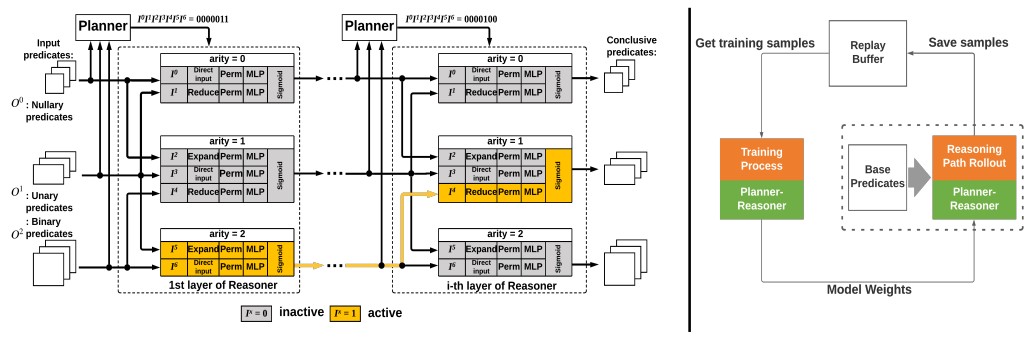

Figure 2: Left (A): The Planner-Reasoner Architecture with an example solution. Right (B): The overall (end-to-end) training process of the model by deep reinforcement learning. "Perm" denotes `Permute` operator.

## 3.1 THE OVERALL MODEL ARCHITECTURE OF PRIMA

Logic reasoning typically seeks to chain an appropriate sequence of logic operators to reach desirable logical consequences from premises. There are two types of chaining to conduct logic reasoning: (i) forward-chaining and (ii) backward-chaining. The forward-chaining strategy recursively deduces all the possible conclusions based on all the available facts and deduction rules until it reaches the answer. In contrast, backward-chaining starts from the desired conclusion (i.e., the goal) and then works backward recursively to validate conditions through available facts. This paper adopts the forward-chaining strategy since the goal is unavailable here beforehand. Note that, in the MTR setting, it generally requires a large set of "skills" (logic operators) to tackle diverse tasks. Therefore, it is unrealistic to recursively generate all possible conclusions by chaining all the available deduction rules as in the vanilla forward-chaining strategy. Therefore, pruning the forward-chaining process is essential to improve the reasoning efficiency in the MTR setting.

We note that reasoning via pruned forward-chaining could be implemented by: (i) constructing the elementary logic operators and (ii) selecting and chaining a subset of these logic operators together. Accordingly, we develop a novel neural model architecture, named Planner-Reasoner (see Figure 2A), to jointly accomplish these two missions with two interacting neural modules. Specifically, a "Reasoner" defines a set of neural-logic operators (representing learnable Horn clauses (Horn, 1951)). Meanwhile, a "Planner" learns to select and chain a small subset of these logic operators together towards the correct solution (e.g., the orange-colored path in Figure 2A). The inputs to the Planner-Reasoner model are the base predicates $\{p, p(x), p(x, y), \ldots\}$ that describe the premises. In particular, in the MTR setting, the nullary predicate $p$ characterizes the current task to be solved. This design allows the Planner to condition its decisions on the input task type. We feed the outputs (i.e., the conclusion predicates) into multiple output heads, one for each task (just as in standard multitask learning), to generate predictions. The entire model is trained in a complete data-driven end-to-end approach via deep reinforcement learning (Figure 2B and Section 4): it trains the Planner to dynamically prune the forward-chaining process while jointly learning the logic operators in the Reasoner, leading to a state-of-the-art balance between MTR capability and efficiency (Section 5).

## 3.2 REASONER: TRANSFORMING LOGIC RULES INTO NEURAL OPERATORS

The Reasoner module conducts logical deduction using a set of neural operators constructed from first-order logic rules (more specifically, a set of "learnable" Horn clauses). Its architecture is inspired by NLM (Dong et al., 2019) (details about the difference can be found in Appendix D). Three logic rules are considered as essential meta-rules: `BooleanLogic`, `Expansion`, and `Reduction`.

$$\text{BooleanLogic:} \quad \text{expression}(x_1, x_2, \cdots, x_r) \rightarrow \hat{p}(x_1, x_2, \cdots, x_r),$$

where `expression` is composed of a combination of Boolean operations (`AND`, `OR`, and `NOT`) and $\hat{p}$ is the *output predicate*. For a given $r$-ary predicate and a given permutation $\psi \in S_n$, we define $p_\psi(x_1, \cdots, x_r) = p(x_{\psi(1)}, \cdots, x_{\psi(r)})$ where $S_n$ is the set of all possible permutations as the arguments to an input predicate. The corresponding neural implementation of `BooleanLogic` is

$\sigma\left(\text{MLP}\left(p_\psi(x_1,\cdots,x_r)\right);\theta\right)$, where $\sigma$ is the sigmoid activation function, MLP refers to a multilayer perceptron, a $\text{Permute}(\cdot)$ neural operator transforms input predicates to $p_\psi(x_1,\cdots,x_r)$, and $\theta$ is the learnable parameter within the model. This is similar to the implicit Horn clause with the universal quantifier($\forall$), e.g., $p_1(x) \wedge p_2(x) \to \hat{p}(x)$ implicitly denoting $\forall x\, p_1(x) \wedge p_2(x) \to \hat{p}(x)$. The class of neural operators can be viewed as "learnable" Horn clauses.

`Expansion`, and `Reduction` are two types of meta-rules for quantification that bridge predicates of different arities with logic quantifiers ($\forall$ and $\exists$). `Expansion` introduces a new and distinct variable $x_{r+1}$ for a set of $r$-ary predicates with the universal quantifier($\forall$). For this reason, `Expansion` creates a new predicate $q$ from $p$.

$$\text{Expansion}: \quad p(x_1,x_2,\cdots,x_r) \to \forall x_{r+1}, q(x_1,x_2,\cdots,x_r,x_{r+1}),$$

where $x_{r+1} \notin \{x_i\}_{i=1}^r$. The corresponding neural implementation of `Expansion`, denoted by $\text{Expand}(\cdot)$, expands the $r$-ary predicates into the $(r+1)$-ary predicates by repeating the $r$-ary predicates and stacking them in a new dimension. Conversely, `Reduction` removes the variable $x_{r+1}$ in a set of $(r+1)$-ary predicates via the quantifiers of $\forall$ or $\exists$.

$$\text{Reduction}: \quad \forall x_{r+1}\, p(x_1,x_2,\cdots,x_r,x_{r+1}) \to q(x_1,x_2,\cdots,x_r),\, \text{or}$$
$$\exists x_{r+1}\, p(x_1,x_2,\cdots,x_r,x_{r+1}) \to q(x_1,x_2,\cdots,x_r).$$

The corresponding neural implementation of `Reduction`, denoted by $\text{Reduce}(\cdot)$, reducing the $(r+1)$-ary predicates into the $r$-ary predicates by taking the minimum (resp. maximum) along the dimension of $x_{r+1}$ due to the universal quantifier $\forall$ (resp. existential quantifier $\exists$).

### 3.3 PLANNER: ACTIVATING AND FORWARD-CHAINING LEARNABLE HORN CLAUSES

The Planner is our key module to address the capability-efficiency tradeoff in the MTR problem; it is responsible for activating the neural operators in the Reasoner and chaining them into reasoning paths. Existing learning-to-reason approaches, which are often based on inductive logic programming (ILP) (Cropper et al., 2020; Cropper & Dumancic, 2020; Muggleton & De Raedt, 1994) and the correspondingly neural-ILP methods (Dong et al., 2019; Shi et al., 2020). Conventional ILP methods suffer from several drawbacks, such as heavy reliance on human-crafted templates and sensitivity to noise. On the other hand, neural-ILP methods (Dong et al., 2019; Shi et al., 2020), leveraging the strength of deep learning and ILP, such as the Neural Logic Machine (NLM) (Dong et al., 2019), lack explicitness in the learning process of searching for reasoning paths. Let us take the learning process of NLM for example, which follows an intuitive two-step procedure. It first fully connects all the neural blocks and then searches all possible connections (corresponding to all possible predicate candidates) exhaustively to identify the desired reasoning path (corresponding to the desired predicate).

By using our proposed Planner module, we can strike a better capability-efficiency tradeoff. Rather than conducting an exhaustive search over all possible predicate candidates as in NLM, the Planner prunes all the unnecessary operators and identifies an essential reasoning path with low complexity for a given problem. Consider the following example. As shown in Fig. 1 and Fig. 2A (the highlighted orange-colored blocks), at each reasoning step, the Planner takes the input predicates and determines which neural operators should be activated. The decision is represented as a binary vector — $[I^0 \ldots I^6]$ (termed *operator footprint*) — that corresponds to the neural operators of Expand, Reduce and DirectInput[1] at different arity groups. By chaining these sequential decisions, a sequence of operator footprints is formulated as a reasoning path, as the highlighted orange-colored path in Fig. 2A. Generally, the neural operators defined in the Reasoner can also be viewed as (learnable) Horn Clauses (Horn, 1951; Chandra & Harel, 1985), and the Planner *forward-chains* them into a reasoning path.

## 4 LEARNING-TO-REASON VIA REINFORCEMENT LEARNING

In our problem, the decision variables of the Planner are binary indicators of whether a neural operator module should be activated. Readers familiar with Markov decision processes (MDPs) (Puterman,

---

[1]DirectInput is an identity mapping of the inputs where its following operators of Permute, MLP and sigmoid nonlinearity can directly approximate the `BooleanLogic`.

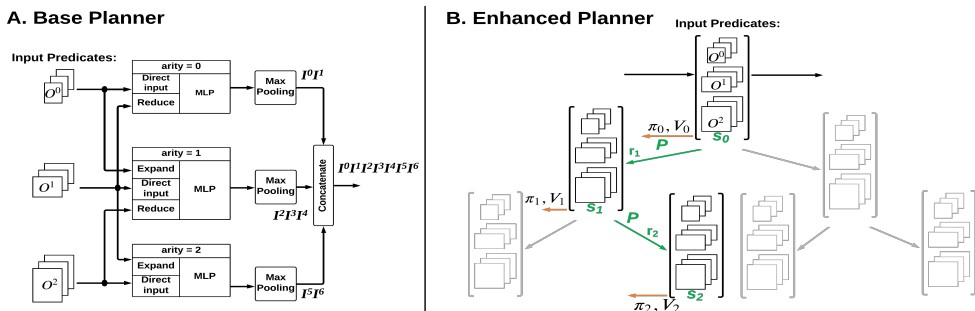

Figure 3: The architecture of the Base Planner and its enhanced version. Left: The Base Planner is based on a one-layer fully-activated Reasoner followed by max-pooling operations to predict an indicator of whether an op should be activated. Right: The Enhanced Planner uses MCTS to further boost the performance.

2014) might notice that the reasoning path of our model in Fig. 2A resembles a temporal rollout in MDP formulation (Sutton & Barto, 2018). Therefore, we frame learning-to-reason as a sequential decision-making problem and adopt off-the-shelf reinforcement learning (RL) algorithms.

### 4.1 AN MDP FORMULATION OF THE LEARNING-TO-REASON PROBLEM

An MDP is defined as the tuple $(\mathcal{S}, \mathcal{A}, P^a_{ss'}, R, \gamma)$, where $\mathcal{S}$ and $\mathcal{A}$ are finite sets of states and actions, the transition kernel $P^a_{ss'}$ specifies the probability of transition from state $s \in \mathcal{S}$ to state $s' \in \mathcal{S}$ by taking action $a \in \mathcal{A}$, $R(s, a) : \mathcal{S} \times \mathcal{A} \to \mathbb{R}$ is the reward function, and $0 \leq \gamma \leq 1$ is a discount factor. A stationary policy $\pi : \mathcal{S} \times \mathcal{A} \to [0, 1]$ is a probabilistic mapping from states to actions. The primary objective of an RL algorithm is to identify a near-optimal policy $\pi^*$ that satisfies

$$\pi^* := \arg\max_{\pi} \left\{ J(\pi) := \mathbb{E}\left[ \sum_{t=0}^{T_{\max}-1} \gamma^t r(s_t, a_t \sim \pi) \right] \right\},$$

where $T_{\max}$ is a positive integer denoting the horizon length — that is, the maximum length of a rollout. The resemblance between Fig. 2 and an MDP formulation is relatively intuitive as summarized in Table 1. At the $t$-th time step, the state $s_t$ corresponds to the set of predicates, $s_t = [O^0_t, O^1_t, O^2_t]$, with the superscript denoting the corresponding arity. The action $a_t = [I^0_t, I^1_t, \ldots, I^{K-1}_t]$ (e.g., $a_0 = [000011]$) is a binary vector that indicates the activated neural operators (i.e., the operator footprint), where $K$ is the number of operators per layer. The reward $r_t$ is defined to be

$$r_t := \begin{cases} -\sum_{i=0}^{K-1} I^i_t, & \text{if } t < T_{\max} \\ \texttt{Accuracy}, & t = T_{\max} \end{cases}. \tag{1}$$

That is, the terminal reward is set to be the reasoning accuracy at the end of the reasoning path (see Appendix A.2 for its definition), and the intermediate reward at each step is chosen to be the negated number of activated operators (which penalizes the cost of performing the current reasoning step). The transition kernel $P^a_{ss'}$ corresponds to the function modeled by one Reasoner layer; each Reasoner layer will take the state (predicates) $s_{t-1} = [O^0_{t-1}, O^1_{t-1}, O^2_{t-1},]$ and the action (operator footprint) $a_t = [I^0_t, I^1_t, \ldots, I^{K-1}_t]$ as its input and then generate the next state (predicates) $s_t = [O^0_t, O^1_t, O^2_t,]$. This also implies that the Reasoner layer defines a deterministic transition kernel, i.e., given $s_{t-1}$ and $a_t$ the next state $s_t$ is determined.

Table 1: The identification between the concepts of PRIMA and that of RL at the $t$-th time step.

| RL | State $s_t$ | Action $a_t$ | Reward $r_t$ | Transition Kernel $P^a_{ss'}$ | Policy | Rollout |
|---|---|---|---|---|---|---|
| PRIMA | Predicates of different arities: $[O^0_t, O^1_t, O^2_t]_t$ | Operator footprint: $[I^0_t \ldots I^{K-1}_t]$ | Eq. (1) | One layer of **Reasoner** | **Planner** | Reasoning path |

### 4.2 POLICY NETWORK: MODELING OF PLANNER

The Planner module is embodied in the policy network. As shown in Fig. 3A, the base planner is a separate module that has the same architecture of 1-layer (fully-activated) Reasoner followed by a max-pooling layer. This architecture enables the reduction of input predicates to the specific indicators by reflecting whether the operations at the corresponding position of one layer of Reasoner are active or inactive. Further, we can also leverage Monte-Carlo Tree Search (MCTS) (Browne et al., 2012; Munos, 2014) to boost the performance, which leads to an Enhanced Planner (Fig. 3B). An MCTS algorithm such as the Upper Confidence Bound for Trees (UCT) method (Kocsis et al., 2006), is a model-based RL algorithm that plans the best action at each time step (Browne et al., 2012) by constructing a search tree, with states as nodes and actions as edges. The Enhanced Planner uses MCTS to exploit partial knowledge of the problem structure (i.e., the deterministic transition kernel $P_{ss'}^a$ defined by the Reasoner layer) and construct a search tree to help identify the best actions (which ops to activate). Details of the MCTS algorithms used in the Enhanced Planner can be found in Appendix A.1.

### 4.3 OVERALL LEARNING FRAMEWORK

As illustrated in Fig. 3, we introduce concrete data-driven decision-making methods—that is, RL approaches (Sutton & Barto, 2018)—to address the learning-to-reason problem. To illustrate this, we apply the model-free RL method REINFORCE (Williams, 1992) and the model-based method MuZero (Schrittwieser et al., 2019). Compared with model-free reinforcement learning, model-based reinforcement learning (MBRL) more effectively handles large search-space problems, such as the game of Go (Silver et al., 2017b,b,a). MuZero (Schrittwieser et al., 2019), a recently proposed MBRL approach to integrating planning and learning, has achieved great success with a variety of complex tasks. Motivated by the success of MuZero, we propose an MBRL approach for neural-symbolic reasoning. The key insight behind adopting MuZero is that in real applications, we typically have partial structural knowledge of the transition kernel $P_{ss'}^a$ and reward function $r(s, a)$. As a result of the model-based module, testing complexity can be greatly reduced by adopting MCTS, which leads to a better set of predicates. Of course, MuZero is just one option in the model-based family of approaches. We leave it as future research to propose and compare other model-based alternatives.

The pipeline of the training process is illustrated in Fig 2.B (the right subfigure). After loading the model weights, the reasoning path rollouts are executed by the agent (or model instance), according to the current policy network. The performed reasoning path rollout is then stored in the replay buffer. The Planner-Reasoner is trained via rollouts sampled from the replay buffer.

## 5 EXPERIMENTAL RESULTS AND ANALYSIS

In this section, we evaluate the performance of different variants of PRIMA on eight tasks from the family tree and graph benchmarks (Graves et al., 2016), including `1-Outdegree`, `AdjacentToRed`, `HasFather`, `HasSister`, `4-Connectivity`, `IsGrandparent`, `IsUncle`, `IsMGUncle`. These tasks are widely used benchmarks for inductive logic programming (Krötzsch, 2020; Calautti et al., 2015). Detailed descriptions about those tasks can be found in Appendix B.1. We evaluate their testing accuracy and reasoning cost (measured in FLOPs: the number of floating-point operations executed (Clark et al., 2020)) on these tasks and compare them to several baselines. Furthermore, detailed case studies are conducted on the reasoning path, which indicates the operator sharing among different tasks. All the results demonstrate the graceful capability-efficiency tradeoff of PRIMA in multi-task reasoning.

### 5.1 EXPERIMENTAL SETUPS

In the multi-task setting, we first randomly sample a task according to a pre-defined probability distribution (over different tasks), and then generate the data for the selected task using the same methods as in NLM (Dong et al., 2019). In addition, we also augment the generated data with one-hot encoding to indicate the selected task, which will be further converted into nullary (background) input predicates. Also, task-specific output heads are introduced to generate outputs for different tasks. These adaptions apply to NLM-MTR, DLM-MTR, and PRIMA. The reasoning accuracy is

Table 2: **Testing Accuracy and PSS** of different variants of PRISA on different tasks. PRISA-MuZero achieves the best performance on single-task reasoning, which confirms the strength of the MCTS-based Enhanced Planner and the MuZero learning strategy. "*m*": the problem size. "PSS": Percentage of Successful Seeds.

| | | | Family Tree | HasFather | HasSister | IsGrandparent | IsUncle | IsMGUncle | Graph | AdjacentToRed | 4-Connectivity | 1-OutDegree |
|---|---|---|---|---|---|---|---|---|---|---|---|---|
| testing acc | Single Task | PRISA-REINFORCE | m=20 | 62.6 | 50.7 | 96.5 | 97.3 | 99.8 | m=10 | 47.7 | 33.5 | 48.7 |
| | | | m=100 | 87.8 | 69.8 | 2.3 | 97.7 | 98.4 | m=50 | 71.6 | 92.8 | 97.4 |
| | | | PSS | 0 | 0 | 0 | 0 | 0 | PSS | 0 | 0 | 0 |
| | | PRISA-PPO | m=20 | 71.5 | 64.3 | 97.5 | 98.1 | 99.6 | m=10 | 62.3 | 57.8 | 61.6 |
| | | | m=100 | 93.2 | 78.7 | 98.2 | 97.3 | 99.1 | m=50 | 85.5 | 95.2 | 96.3 |
| | | | PSS | 0 | 0 | 0 | 0 | 0 | PSS | 0 | 0 | 0 |
| | | PRISA-MuZero | m=20 | **100** | **100** | **100** | **100** | **100** | m=10 | **100** | **100** | **100** |
| | | | m=100 | **100** | **100** | **100** | **100** | **100** | m=50 | **100** | **100** | **100** |
| | | | PSS | **100** | **100** | **100** | **100** | **100** | PSS | **90** | **100** | **100** |

used as the reward. In the inference (or testing) stage, the Reasoner is combined with the learned Base Planner to perform the tasks, instead of an enhanced planner (MCTS), to reduce the extra computation. The problem size for training is always 10 for graph tasks and 20 for family tree tasks across all settings, regardless of the sizes of testing problems.

## 5.2 OVERALL PERFORMANCE

**Single-task reasoning capability** Before we proceed to evaluate the MTR capabilities of PRIMA, we first adapt it to the single-task setting by letting the nullary input predicates to be zero, and then train a separate model for each individual task. We name this single-task version of PRIMA as PRISA (i.e., ***P**lanner-**R**easoner **I**nside a **S**ingle Reasoning **A**gent*), and compare it with existing (single-task) neural-logic reasoning approaches (e.g., NLM (Dong et al., 2019), DLM (Matthieu et al., 2021), and MemNN (Sukhbaatar et al., 2015)). First, we compare the performance of three different variants of PRISA (Section 4.3) for single-task reasoning, which learns their planners based on different reinforcement learning algorithms; PRISA-REINFORCE uses REINFORCE (Williams, 1992), PRISA-PPO uses PPO (Schulman et al., 2017), and PRISA-MuZero uses MuZero (Schrittwieser et al., 2019). We report the test accuracy and the Percentage of Successful Seeds (PSS) in Table 2 to measure the model's reasoning *capabilities*, where the PSS reaches 100% of success rates (Matthieu et al., 2021). We note that PRISA-MuZero has the same 100% accuracy as NLM (Dong et al., 2019), DLM (Matthieu et al., 2021), and ∂ILP (Evans & Grefenstette, 2018) across different tasks, and outperforms MemNN (Sukhbaatar et al., 2015) (shown in the single-task part in Table 3). But it also has a higher successful percentage (PSS) in comparison with other methods. The results show that PRISA-MuZero achieves the best performance on single-task reasoning, which confirms the strength of the MCTS-based Enhanced Planner (Section 4.3) and the MuZero learning strategy. Therefore, we will use the Enhanced Planner and MuZero in PRISA and PRIMA in the rest of our empirical studies.

**Multi-task reasoning capability** Next, we evaluate the MTR capabilities of PRIMA. To the best of our knowledge, there is no existing approach that is designed specifically for MTR. Therefore, we adapt NLM and DLM into their multi-task versions, named NLM-MTR and DLM-MTR, respectively. NLM-MTR and DLM-MTR follow the same input and output modification as what we did to upgrade PRISA to PRIMA (Section 3.1). By this, we can examine the contribution of our proposed *Planner-Reasoner architecture* for MTR. As shown in Table 3, PRIMA (with MuZero as the Base Planner) performs well (perfectly) on different reasoning tasks. On the other hand, DLM-MTR experiences some performance degradation (on `AdjacentToRed`). This result confirms that our Planner-Reasoner architecture is more suitable for MTR. We conjecture that the benefit comes from using a Planner to explicitly select the necessary neural operators for each task, avoiding potential conflicts between different tasks during the learning process.

Experiments are also conducted to test the performance of PRIMA with different problem sizes. The problem size in training is 10 for all graph tasks and 20 for all family-tree tasks. In testing, we evaluate the methods on much larger problem sizes (50 for the graph tasks and 100 for the family tree tasks), which the methods have not seen before. Therefore, the Planner must dynamically activate a proper set of neural operators to construct a path to solve the new problem. As reported in Fig. 4 and Tables 2 and 3, PRIMA can achieve the best accuracy and lower flops when the problem sizes for training and testing are different.

**Reasoning efficiency** To measure the reasoning efficiency of the proposed methods at the inference stage, PRIMA is compared with NLM, MemNN, and NLM-MTR in terms of FLOPs. As shown in

Table 3: **Testing Accuracy and PSS** of PRIMA and other baselines on different reasoning tasks. The results of ∂ILP, NLM, and DLM are merged in one row due to space constraints and are presented in the same order if the results are different. Note that PRIMA's test efficiency is superior to NLM-MTR's as shown in Fig. 4. "*m*": the problem size. "PSS": Percentage of Successful Seeds. Numbers in red denote $< 100\%$.

| | | | Family Tree | HasFather | HasSister | IsGrandparent | IsUncle | IsMGUncle | Graph | AdjacentToRed | 4-Connectivity | 1-OutDegree |
|---|---|---|---|---|---|---|---|---|---|---|---|---|
| testing accuracy | Single Task | MemNN | m=20 | 99.9 | 86.3 | 96.5 | 96.3 | 99.7 | m=10 | 95.2 | 92.3 | 99.8 |
| | | | m=100 | 59.8 | 59.8 | 97.7 | 96 | 98.4 | m=50 | 93.1 | 81.3 | 78.6 |
| | | | PSS | 0 | 0 | 0 | 0 | 0 | PSS | 0 | 0 | 0 |
| | | ∂ILP/ NLM/ DLM | m=20 | 100 | 100 | 100 | 100 | 100 | m=10 | 100 | 100 | 100 |
| | | | m=100 | 100 | 100 | 100 | 100 | 100 | m=50 | 100 | 100 | 100 |
| | | | PSS | 100 | 100 | 100 | 100/ 90/ 100 | 100/ 20/ 70 | PSS | 100/ 90/ 90 | 100 | 100 |
| | Multi-Task | NLM-MTR | m=20 | 100 | 100 | 100 | 100 | 100 | m=10 | 100 | 100 | 100 |
| | | | m=100 | 100 | 100 | 100 | 100 | 100 | m=50 | 100 | 100 | 100 |
| | | | PSS | 100 | 100 | 100 | 100 | 90 | PSS | 90 | 100 | 100 |
| | | DLM-MTR | m=20 | 100 | 100 | 100 | 100 | 100 | m=10 | 96.7 | 100 | 100 |
| | | | m=100 | 100 | 100 | 100 | 100 | 100 | m=50 | 97.2 | 100 | 100 |
| | | | PSS | 100 | 100 | 100 | 100 | 100 | PSS | 0 | 100 | 100 |
| | | PRIMA | m=20 | 100 | 100 | 100 | 100 | 100 | m=10 | 100 | 100 | 100 |
| | | | m=100 | 100 | 100 | 100 | 100 | 100 | m=50 | 100 | 100 | 100 |
| | | | PSS | 100 | 100 | 100 | 100 | 90 | PSS | 90 | 100 | 100 |

Fig. 4, NLM and NLM-MTR demonstrate a similar performance and suffer the highest reasoning cost when it is tested with large problem sizes, such as 50 for graph tasks and 100 for family tree tasks. For MemNN, although the FLOPs of it seem low in most cases of testing, its testing accuracy is bad and cannot achieve accurate predictions (Table 3). In contrast, PRIMA can significantly reduce the reasoning complexity by intelligently selecting ops using a planner. Overall, PRIMA strikes a satisfactory capability-efficiency tradeoff in comparison with all available multi-tasking baselines.

**Operator/Path sharing in MTR**   To take a closer look into how PRIMA achieves such a better capability-efficiency tradeoff (in Fig. 4), we examine the reasoning paths on three different graph tasks: `1-Outdegree`, `AdjacentToRed`, and `4-Connectivity`. Specifically, we sample instances from these three tasks and feed them into PRIMA separately to generate their corresponding reasoning paths. The results are plotted in Fig. 5A, where the gray paths denote the ones shared across tasks, and the colored ones are task-specific. It clearly shows that PRIMA learns a large set of neural operators sharable across tasks. Given each input instance from a particular task, PRIMA activates a set of shared paths along with a few task-specific paths to deduce the logical consequences.

**Generalizability of the Planner**   To demonstrate the generalizability of the Planner module in PRIMA, we generate input instances of `AdjacentToRed` with topologies that have not been seen during training. In Fig. 5B, we show the reasoning paths activated by the Planner for these different topologies, which demonstrates that different input instances share a large portion of reasoning paths. This fact is not surprising as solving the same task of `AdjacentToRed` should rely on a common set of skills. More interestingly, we notice that even for solving this same task, the Planner will have to call upon a few *instance-dependent* sub-paths to handle the subtle inherent differences (e.g., the graph topology) that exist between different input instances. For this reason, PRIMA maintains an instance-dependent *dynamic* architecture, which is in sharp contrast to the Neural Architecture Search (NAS) approaches. Although NAS may also use RL algorithms to seek for a smaller architecture (Zoph & Le, 2017), it only searches for a *static* architecture that will be applied to all the input instances.

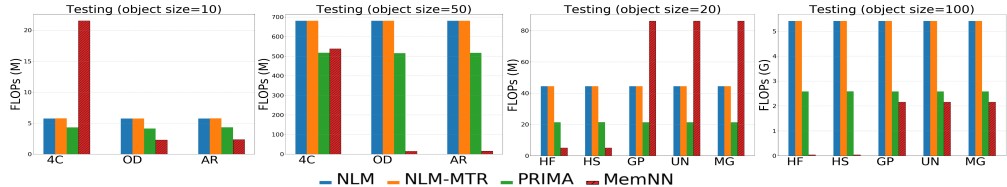

Figure 4: The reasoning costs (in FLOPs) of different models at the inference stage (1 M=$1 \times 10^6$, 1 G=$1 \times 10^9$). Compared to NLM and NLM-MTR, our PRIMA significantly reduces the reasoning complexity by intelligently selecting ops (short for neural operators) using a planner. Although the FLOPs of MemNN seem low in most cases of testing, its testing accuracy is bad and cannot achieve accurate predictions (see Table 3). "OD" denotes `1-Outdegree`, likewise, "AR":`AdjacentToRed`, "4C":`4-Connectivity`, "HF":`HasFather`, "HS":`HasSister`, "GP":`IsGrandparent`, "UN":`IsUncle`, "MG":`IsMGUncle`.

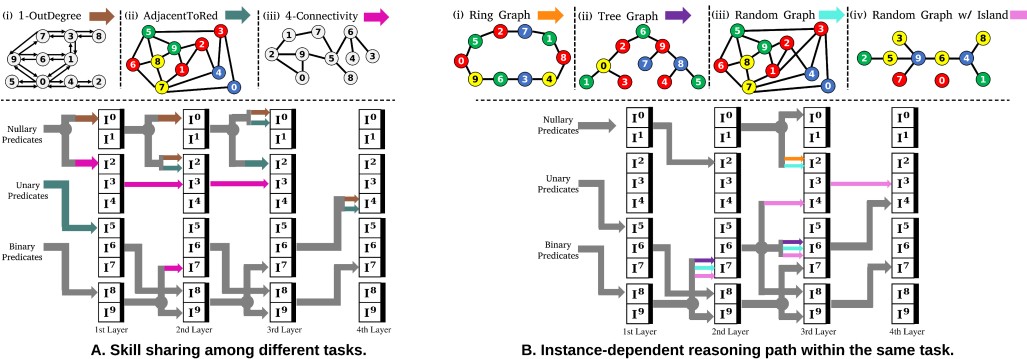

Figure 5: The reasoning paths of PRIMA. The gray arrows denote the "shared path". The colored arrows in sub-figures (A) and (B) denote task-specific paths and instance-specific paths, respectively. In (A), besides the AdjacentToRed task we have introduced earlier, we also consider the 1-Outdegree task, which reasons about whether the out-degree of a node is exactly equal to 1, and the 4-Connectivity task, which is to decide whether there are two nodes connected within 4 hops.

# 6 RELATED WORK

**Multi-task learning**   Multi-task Learning (Zhang & Yang, 2021; Zhou et al., 2011; Pan & Yang, 2009) has focused primarily on the supervised learning paradigm, which itself can be divided into several approaches. The first is feature-based MTL, also called multi-task feature learning, (Argyriou et al., 2008), which explores the sharing of features across different tasks using regularization techniques (Shinohara, 2016; Liu et al., 2017). The second approach assumes that tasks are intrinsically related, such as low-rank learning  (Ando & Zhang, 2005; Zhang et al., 2005), learning with task clustering (Gu et al., 2011; Zhang et al., 2016), and task-relation learning (such as via task similarity, task correlation, or task covariance) (Goncalves et al., 2016; Zhang, 2013; Ciliberto et al., 2015). MTL has also been explored under other paradigms such as unsupervised learning, for example, via multi-task clustering in (Zhang & Zhang, 2013; Zhang et al., 2015b; Gu et al., 2011; Zhang, 2015).

**Neural-symbolic reasoning**   Neural-symbolic AI for reasoning and inference has a long history (Besold et al., 2017; Bader et al., 2004; Garcez et al., 2008), and neural-ILP has developed primarily in two major directions  (Cropper et al., 2020). The first direction applies neural operators such as tensor calculus to simulate logic reasoning (Yang et al., 2017; Dong et al., 2019; Shi et al., 2020) and (Manhaeve et al., 2018). The second direction involves relaxed subset selection (Evans & Grefenstette, 2018; Si et al., 2019) with a predefined set of task-specific logic clauses. This approach reduces the task to a subset-selection problem by selecting a subset of clauses from the predefined set and using neural networks to search for a relaxed solution.

Our work is different from the most recent work, such as (Dong et al., 2019; Shi et al., 2020), in several ways. Compared with (Dong et al., 2019), the most notable difference is the improvement in efficiency by introducing learning-to-reason via reinforcement learning. A second major difference is that PRIMA offers more generalizability than NLM by decomposing the logic operators into more finely-grained units. We refer readers for more detailed related work to Appendix D.

# 7 CONCLUSION

A long-standing challenge in multi-task reasoning (MTR) is the intrinsic conflict between capability and efficiency. Our main contribution is the development of a novel neural-logic architecture termed PRIMA, which learns to perform efficient multitask reasoning (in first-order logic) by dynamically chaining learnable Horn clauses (represented by the neural logic operators). PRIMA improves inference efficiency by dynamically pruning unnecessary neural logic operators, and achieves a state-of-the-art balance between MTR capability and inference efficiency. The model's training follows a complete data-driven end-to-end approach via deep reinforcement learning, and the performance is validated across a variety of benchmarks. Future work could include extending the framework to high-order logic and investigating scenarios when meta-rules have a hierarchical structure.

## REPRODUCIBILITY STATEMENT

We commit to ensuring that other researchers with reasonable background knowledge in our area can reproduce our theoretical and empirical results. The algorithm details can be seen in Appendix A. Specifically, benchmark details are provided in Appendix B.1, hyper-parameter settings are provided in Appendix B.2, and computing infrastructure are detailed in Appendix B.3. The experiment details can be seen in Appendix B.

## ETHICS STATEMENT

This work is about the methodology of achieving a capability-efficiency trade-off in multi-task first-order logic reasoning. The potential impact of this work is likely to further extend the framework to high-order logic and investigate scenarios when meta-rules have a hierarchical structure, which should be generally beneficial to the multi-task learning research community. We have not considered specific applications or practical scenarios as the goal of this work. Hence, it does not have any direct ethical consequences.

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
