# OpenReview forum: "PRIMA: Planner-Reasoner Inside a Multi-task Reasoning Agent"
_ICLR.cc/2022/Conference — ICLR 2022 Submitted_

### Official Review · Reviewer_Z6sN · 2021-10-26

**Correctness:** 3
**Technical Novelty And Significance:** 3
**Empirical Novelty And Significance:** 3
**Recommendation:** 6
**Confidence:** 3

**Main Review:**

The article is interesting, but in my opinion, badly structured. It is unclear what new features have been introduced. In fact, the title and introduction only refer to PRIMA, while the article spends a lot of space presenting PRISA (the single-task version) as well. The experimental section is also divided equally between the two systems, making the purpose of the article unclear.
I would recommend reformulating the title and the introductory parts to make them fit better with the content of the article, which is not only about PRIMA but also about PRISA. The differences between the two systems are not many, since PRIMA is based on PRISA, but in my opinion, the organisation of the content and the title leave the reader a bit lost.

In addition, the article should be re-read to correct some grammatical errors. For example, on page 1 "to solving multiple tasks". Also on this page is a sentence disconnected from the rest of the text: "Existing MTL models." A pagina 6, "to integrating". On page 8 "we examine its the reasoning paths".

**Summary Of The Paper:**

The paper presents PRIMA, a framework designed to solve the problem of multi-task reasoning. PRIMA consists of two parts: a reasoner, which produces logical deductions using a set of neural operators built by translating FOL rules; and a planner, which activates the reasoner operators and creates reasoning paths to solve the task.
The paper presents several tests that achieve performance usually stable on 100% succesful seeds.

**Summary Of The Review:**

The system is able to achieve really good results in the tests conducted. On the other hand, the article is not structured very well and not entirely consistent with the title.

---

### Official Review · Reviewer_eedL · 2021-11-03

**Correctness:** 4
**Technical Novelty And Significance:** 4
**Empirical Novelty And Significance:** 3
**Recommendation:** 6
**Confidence:** 3

**Main Review:**

Learning to improve reasoning is an interesting and overlooked problem, particularly in purely learning-based systems, though they do have a long history in more traditional reasoning circles (for example the authors may want to look up utility problems in search-based reasoning frameworks [1]). I think the use of RL methods to direct reasoning is an interesting solution. In terms of concerns regarding the work, I mostly feel the need to improve reasoning efficiency was not as well-motivated and there is missing information on additional training overhead introduced by including an RL component to the overall learning problem.

Reasoning Cost for Single Task Problem: The paper motivates the problem of improving reasoning efficiency by appealing to MTR problems. While I agree that achieving efficient reasoning may be harder in MTR settings due to the need to generalize across more diverse tasks, I don’t see how the same is not a problem in single task settings. I would imagine PRISA presents the same gains in improvements over vanilla NLM that is presented by PRIMA over NLM-MTR. For me, the problem of introducing a planner-reasoner into an NLM style framework seems independent of the problem of extending NLM into a multi-task learning setting. So I would encourage the authors to include computational costs for PRISA in the results too. If it does turn out that PRIMA somehow is more efficient than PRISA, that might be worth pointing out too.

Training Overhead: In terms of computational cost, the paper exclusively focuses on the cost during inference. Unfortunately, this doesn’t present the complete picture. By introducing a planner-reasoner into the mix, the authors are currently looking at a more complex learning problem. Any gains at the time of reasoning are achieved at the cost of performing trials at the training time that tries to identify policies that promote smaller operator footprints. I would imagine, this introduces additional computational costs (not to mention those introduced by the use of MCTS). Unless the authors are explicitly focusing on specific application settings where the computational cost during training can be ignored, the authors should provide a discussion on the overhead introduced during training (and may be provide a comparison of training computational costs). After all, most concerns regarding energy usage of modern deep learning systems are related to the training time. If the authors do have some application setting in mind then there should be a clearer discussion of the setting in the paper.

Smaller Comments:

The description of neural implementation of BooleanLogic in section 2.2 seems to be incomplete. The corresponding text refers to the Permute operator which is missing from the description.

Some of the claims in the empirical evaluation need to be more carefully worded. For example,

“We note that PRISA-MuZero has the same 100% accuracy as NLM  and \deltaILP across different tasks” -- doesn’t  \deltaILP outperform PRISA-MuZero in AdjacentToRed task?

“On the other hand, DLM-MTR experiences some performance degradation (on AdjacentToRed). This result confirms that our Planner-Reasoner architecture is more suitable for MTR.” -- But this statement overlooks the fact that NLM-MTR does as well as PRIMA for all the tasks. The second statement (and the following statement) makes it sound like planner-reasoner provides an advantage over all the baselines that don’t use a Planner-Reasoner architecture.

Reward Function for RL: Currently there is no hyper-parameter introduced to balance the penalty related to the operator footprint size against the reward-related to the final accuracy. I expect for larger arities there could be cases in the current setting, the RL algorithm could choose lower accuracy to avoid larger operator footprints.

[1] Minton, Steven. "Quantitative results concerning the utility of explanation-based learning." Artificial Intelligence 42.2-3 (1990): 363-391.

**Summary Of The Paper:**

The paper introduces a neuro-symbolic framework for multi-task reasoning that builds on NLM. Specifically, the method tries to achieve higher efficiency at the inference time, by using an RL component that controls the operations performed as part of the reasoning process. They adapt the system for multi-task settings by introducing nullary predicates that capture the specific task details. The method is compared to multiple baselines, including an adaptation of NLM to the multi-task setting. While the methods perform similar to the baselines in terms of the overall accuracy, the approach does result in a lower number of floating-point operations at the inference time.

**Summary Of The Review:**

I think the paper looks at an often overlooked problem and introduces an interesting mechanism to drive the reasoning process. Though I do have concerns about some experimental results that are missing and could provide more information about trade-offs introduced by including a planner-reasoner in the architecture.

---

### Official Review · Reviewer_Phsj · 2021-11-03

**Correctness:** 3
**Technical Novelty And Significance:** 3
**Empirical Novelty And Significance:** 2
**Recommendation:** 6
**Confidence:** 3

**Main Review:**

In general, PRIMA seems like a strong approach for multi-task logical reasoning. While the paper motivates PRIMA using the general framing of the generalization/single-task performance trade-off in multi-task learning more generally, I find this a bit tenuous (not a reason to reject, but I recommend the authors maybe temper their claims a bit). I think multi-task logical reasoning is an important problem in itself, without the broader connection (as it’s hard to even think how we’d get an approach like this to generalize to say multi-task problems in NLP/Vision/Robotics).

Strengths:

In terms of strengths, I think the approach is well-formulated, and easy to understand. Leveraging th structure of the problem to do MCTS and train a MuZero style policy (planner) is a great idea, and one that clearly paid dividends. I think the components of the proposed architecture were well motivated and well executed.

Weaknesses:

That being said, I have identified a few weaknesses, and I am hoping the authors can either point me to relevant information (I totally could have misunderstood certain results), or engage with me to discuss the importance of these few points:

As it stands, we evaluate PRIMA in both multi-task and single-task settings, on a set of logical reasoning tasks on family trees/graphs. These tasks, while motivated in prior work, feel very simple to me, further evidenced by the fact that PRIMA, as well as most prior work get 100% accuracy across the board, except for a handful of tasks. I’m wondering if we can trust this evaluation to truly get at the broad capabilities of this model, relative to those that already exist.

The key “win” for PRIMA over related work seems to be the reduction in FLOPs relative to prior methods like MT-NLM, MT-DLM (variants of prior logical reasoning models augmented with the “simple” task conditioning information). Performance is the same. However, it’s hard for me to understand (1) if this reduction of FLOPs actually matters in practice, (2) if this is a fair comparison (I don’t know these prior models too well, but they could be using more FLOPs for expressivity, because they considered compute “cheap” for these settings), and (3) if the corresponding reduction is meaningfully significant (e.g., it’s a 2X reduction, but is it a 2X reduction that just translates to an extra second on a laptop, or an extra few hours on a huge cluster).

The second “win” is a single probing experiment done towards the end of the paper, where PRIMA (end of Section 4) is fed new input instances with novel topologies for a graph reasoning problem. The argument here, mostly graphical, is hard to follow, but seems to indicate that PRIMA has a bias for reusing reasoning components when possible, but uses instance=dependent sub-paths when dealing with novel features. I’m not really sure how to interpret this argument, nor am I sure whether this sort of functionality is present in the baselines.

Typos/Style/Questions:

Page 1: “Existing MTL models.” (fragment)

---
EDIT (Post Rebuttal): The authors have addressed some (but not all) of my weaknesses in a satisfactory way. I am raising my score up to a 6.

**Summary Of The Paper:**

This paper presents PRIMA (“Planner-Reasoner Inside a Multi-Task Reasoning Agent”), a multi-task reasoning model that can be applied to several tasks that require first order logical reasoning. Notably, this paper seeks to address a problem that plagues multi-task learning more generally — the tradeoff between cross-task generalizability, and single-task effectiveness. Prior work in multi-task learning shows that it’s hard to strike this balance, and most work in building neural systems for logical reasoning focus on the single task setting.

The main contribution of this paper is first a single task planning-based agent that learns to reason for an individual logical task (e.g., a graph reasoning task like `4-Connectivity` or `1-Outdegree`) by routing input problem representations (predicates describing the problem) through a planning module; this planner acts as a high-level policy that (over multiple steps) routes the input through individual “reasoning” components that are “neural” versions of first-order logic meta-operators. This is a compact, two-party approach for solving complex reasoning problems; the reasoner defines the atomic operations, while the planner figures out what operations to apply, in what order.

The full system is learned via reinforcement learning, first via model-free methods such as REINFORCE and PPO, but with significantly better results using the structure of the problem (we can always roll-out through the planner) to use Model-based RL, specifically the MuZero formulation. Using MuZero/MCTS to learn the Planner/Reasoner, PRIMA shows competitive performance on single-tasks, and to generalize to multiple tasks, we simply add a “nullary predicate” that specifies the task ID. PRIMA performs well, comparably to other approaches for logical reasoning, even when prior approaches (such as the Neural Logic Machine) are extended to multi-task settings in the same way (adding this extra task conditioning information). Where PRIMA shows benefits over prior approaches are in aggregate FLOPs (showing that PRIMA may be learning to reason more efficiently), and in a single probing experiment showing how PRIMA can generalize to new input classes unseen at training.

**Summary Of The Review:**

In general, PRIMA is a neat approach for multi-task logical reasoning. However, while the method and optimization are sound, I’m slightly concerned with the results. The two big wins for PRIMA over prior methods are seemingly a reduction in FLOPs (which is hard to contextualize), and possibly evidence that indicates that PRIMA uses shared structure (which has no comparison to baselines). I think this could be a strong paper, and I welcome responses from the authors during rebuttal; I’d be happy to change my score if the above questions are meaningfully addressed.

---

### Official Review · Reviewer_z1ve · 2021-11-07

**Correctness:** 2
**Technical Novelty And Significance:** 2
**Empirical Novelty And Significance:** 2
**Recommendation:** 5
**Confidence:** 3

**Main Review:**

The strength of the paper is a motivating description of the problem, detailed presentation of approach, evaluation and supplementary material.

The problems with the paper are that:
a)  it pays cursory attention to important details about planning, the main idea, of the approach. Specifically, the field of planning is diverse with model-based (PDDL-based) and model-free RL approaches. The paper is only referring to the latter but makes no mention of it.

b) glosses that planning is not just chaining. The complexity of planning increases when actions interact with each other and there is increase in the "type of actions". The authors do not talk about diversity in type of actions and their interaction. If there is no interaction among actions, any chain will give us the solution. The trade-off between different search paths is just cost, not feasibility of finding a plan -- a simpler problem.

c) does not talk how the different tasks relate to each other in evaluation. The ones of family relationship (father, sister, grandparent, uncle and mguncle) seem quite similar.

d) does not discuss the implication of results. The approach is a rather heavy-weight machinery to use on such simple set of reasoning tasks.


**Summary Of The Paper:**

The paper claims to present a new approach for multi-task reasoning where reasoning operators are defined as neural operators and a planner is used to chain them based on tasks. The approach is end-to-end trained and it is claimed to show good improvement across the demonstrated tasks.

**Summary Of The Review:**

The paper is motivated on an important problem - MTR. The approach is also on a promising intuition that a planner should be able to chain the logical operators based on the needs to a task. However, the approach is not well explained with respect to related work and planning capability is not well utilized.

Beyond improving explaining, I will suggest the authors to attempt more mainstream reasoning tasks and not on toy settings.

---

### Decision · Program_Chairs · 2022-01-20

**Decision:**

Reject

**Comment:**

Reviews for this paper were mixed (6,6,6,5) with one review (Z6sN) being somewhat uninformative. During the rebuttal, some reviewers raised their scores to 6 but overall there was not strong excitement among the reviewers, AC, and SAC. From fresh readings (by SAC and researchers with relevant expertise), this paper’s technical approach looks reasonable but feels quite incremental (novelty is not high) and the experimental results are conducted on a small scale problem where up to 100% success rate is achievable by baseline methods. Therefore, the practical significance of this approach for real-world problems with complex and noisy environments is quite unclear. Overall, the paper looks below the ICLR acceptance threshold. For improvement, we suggest providing evidence/demonstration that this method can successfully tackle more challenging real-world problems.